



**1    Projected global tropospheric ozone impacts on vegetation under different**
**2    emission and climate scenarios**

Sicard Pierre[1], Anav Alessandro[2], De Marco Alessandra[3], Paoletti Elena[2]
[1] ACRI-HE, Sophia Antipolis, France
[2] Institute of Sustainable Plant Protection, National Research Council, Sesto Fiorentino, Italy
[3] Italian National Agency for New Technologies, Energy and the Environment, C.R. Casaccia, Italy
**Abstract**
The impact of ground-level ozone ($O_3$) on vegetation is largely under-investigated at global
scale despite worldwide large areas are exposed to high surface $O_3$ levels and concentrations
are expected to increase in the next future. To explore future potential impacts of $O_3$ on
vegetation, we compared historical and projected $O_3$ concentrations simulated by six global
atmospheric chemistry transport models on the basis of three representative concentration
pathways emission scenarios (i.e. RCP 2.6, 4.5, 8.5). To assess changes in the potential $O_3$
threat to vegetation, we used the AOT40 metric. Results point out a significant overrun of
AOT40 in comparison with the recommendations of UNECE for the protection of vegetation.
In fact, many areas of the northern hemisphere show that AOT40-based critical levels will be
exceeded by a factor of at least 10 under RCP8.5. Changes in surface $O_3$ by 2100 range from
about + 4-5 ppb worldwide in RCP8.5 scenario to reductions of about 2-10 ppb in the RCP2.6
scenario. The risk of $O_3$ injury for vegetation decreased by 61% and 47% under RCP2.6 and
RCP4.5, respectively and increased by 70% under RCP8.5. Key biodiversity areas in South
and North Asia, central Africa and Northern America were identified as being at risk from
high $O_3$ concentrations. To better evaluate the regional exposure of ecosystems to $O_3$
pollution, we recommend the use of improved chemistry-climate modelling system, fully
coupled with dynamic vegetation models.
* Corresponding author: pierre.sicard@acri-he.fr
**Keywords:** AOT40, Ozone, Representative Concentration Pathways, $O_3$ injury on vegetation



**Introduction**

Tropospheric ozone ($O_3$) is a secondary air pollutant, i.e. it is not emitted as such in the air but it is formed by reactions among precursors (e.g. $CH_4$, VOCs, NOx). Ozone is an important greenhouse gas resulting in a direct radiative forcing of 0.35-0.37 W m$^{-2}$ on climate (Shindell et al., 2009; Ainsworth et al., 2012). Despite significant control efforts and legislation to reduce $O_3$ precursor emissions, tropospheric $O_3$ pollution is still a major air quality issue over large regions of the Globe (Lefohn et al., 2010; Langner et al., 2012; Young et al., 2013; Cooper et al., 2014; EEA, 2015; Sicard et al., 2016a,b). Long-range transport of $O_3$ and its precursors can elevate the local and regional $O_3$ background concentrations (Ellingsen et al., 2008; Wilson et al., 2012; Paoletti et al., 2014; Derwent et al., 2015; Xing et al., 2015; Sicard et al., 2016a). Therefore, remote areas such as the Arctic region, can be affected (Langner et al., 2012). The current tropospheric $O_3$ levels (35-50 ppb in the northern hemisphere, NH) are high enough to damage both forests and crops by reducing growth rates and productivity (Paoletti et al, 2009; Wittig et al., 2009; Anav et al., 2011; Mills et al., 2011; Ashworth et al., 2013; Proietti et al., 2016).

Increasing atmospheric $CO_2$, nitrogen deposition and temperatures enhance plant growth, and increase primary production and greening of plants (Nemani et al., 2003; Zhu et al., 2016). At the global scale, a widespread increase of greening and net primary production (NPP) is observed over 25-50% of the vegetated area, while a decrease is observed over only 7% of the Globe (Nemani et al., 2003; Zhu et al., 2016). In contrast, a previous modeling study over Europe shows how $O_3$ reduces the mean annual gross primary production (GPP) by about 22% and the leaf area index by 15-20% (Anav et al., 2011). Similarly, Proietti et al (2016), using different *in-situ* measurements collected over 37 European forest sites, found a GPP decrease of 30% caused by $O_3$. At global scale, over the time period 1901-2100, GPP is projected to decrease by 14-23% (Sitch et al., 2007). As a consequence of reduced photosynthetic assimilation, the total biomass of trees is estimated to be decreased by 7% under the current $O_3$ mean concentrations (40 ppb) and by 17% under the $O_3$ mean concentrations expected in 2100 (97 ppb) compared to preindustrial $O_3$ levels (about 10 ppb, Wittig et al., 2009). Wittig et al. (2009) also reported that the total tree biomass of angiosperms was reduced by 23% at $O_3$ mean concentrations of 74 ppb, and by 7% at 92 ppb for gymnosperms. High surface $O_3$ levels, exceeding 40 ppb, do occur in many regions of the Globe with associated economic costs of several billion dollars per year (Wang and Mauzerall, 2004; Ashmore, 2005). Ashworth et al. (2013) reported an annual loss of 3.5% for



wheat (very $O_3$-sensitive) and 1% for maize (more $O_3$-tolerant) for Europe in 2010 relative to
2000, while Holland et al. (2006) estimated a €4.5 billion loss in the production of 23
common crop species, due to surface $O_3$ exposure by 2020 relative to 2000.

The international Tropospheric Ozone Assessment Report (TOAR) establishes a state-of-the-
art and an up-to-date scientific assessment of global $O_3$ metrics for climate change, human
health and crop/ecosystem research (Lefohn et al. 2017). To assess the potential $O_3$ risk and
protect vegetation from $O_3$, different metrics are used: the European and US standard (AOT40
and W126, respectively) are based on exposure-based metrics, while flux-based metrics have
been introduced only recently (UNECE, 2010; Klingberg et al., 2014; EEA, 2015). Unlike the
exposure-based metrics, which only rely on the surface $O_3$ concentration, the flux-based
metrics were developed to quantify the accumulation of damaging $O_3$ taken up by vegetation
through the stomata over a species-specific phenological time-window. These metrics also
provide an information-rich tool in assessing the relative effectiveness of air pollution control
strategies in lowering surface $O_3$ levels worldwide (Monks et al., 2015). By reducing plant
photosynthesis and growth, high tropospheric $O_3$ levels will result in reduction in carbon
storage by vegetation and, *in fine* an indirect radiative forcing as a consequence of the $CO_2$
rising in the atmosphere (Sitch et al., 2007; Ainsworth et al., 2012). This $CO_2$ rising reduces
stomatal conductance which decreases $O_3$ flux into plants leading to increased $O_3$ levels in the
air of 3-4 ppb during the growing season over the NH by doubling of $CO_2$ concentration
(Fiscus et al., 2005; Sanderson et al., 2007).

Projected changes in tropospheric $O_3$ vary considerably among models (Stevenson et al.,
2006; Wild, 2007) and emission scenarios. In earlier studies, the emissions of $O_3$ precursors
were based on a high population growth, leading to very high projected surface $O_3$
concentrations by 2100 (Stevenson et al., 2000; Zeng and Pyle, 2003; Shindell et al., 2006).
The last emission scenarios, i.e. the Representative Concentration Pathways (RCPs) were
developed as part of the Fifth Assessment Report of the Intergovernmental Panel on Climate
Change (Meinshausen et al., 2011; van Vuuren et al., 2011; Cubasch et al., 2013; Myhre et
al., 2013). These scenarios include e.g. different assumptions on climate, energy access
policies, and land cover and land use changes (Arneth et al., 2008; Kawase et al., 2011;
Kirtman et al., 2013). Until now, studies on $O_3$ pollution impacts on terrestrial ecosystems are
either limited to a single model or to particular regions (e.g. Clifton et al., 2014; Rieder et al.,
2015) and only a few applications of global or regional models under the new RCPs scenarios



were carried out (Kelly et al., 2012). In the framework of the Atmospheric Chemistry and
Climate Model Intercomparison Project (ACCMIP), different simulations were performed by
Lamarque et al. (2013) and Young et al. (2013) from 16 global chemistry models.

A few issues about surface $O_3$, such as a better understanding of spatial changes and a better
assessment of $O_3$ impacts worldwide, are still challenging. To overcome these issues, the aim
of this study is to quantify, for the first time, the spatial and temporal changes in the projected
potential $O_3$ impacts on carbon assimilation of vegetation at global scale, by comparing the $O_3$
potential injury at present with that expected at the end of the 21$^{st}$ century from different
global chemistry models.

**Materials and Methods**
*ACCMIP models and RCP scenarios*
The global chemistry models used in this work have been developed under the ACCMIP
project. A detailed description of the selected models and of the emission scenarios (i.e.
RCPs) is included in Supplementary Information (SI). ACCMIP models have been widely
validated and used to evaluate projected changes in atmospheric chemistry and air quality
under different emission and climate assumptions (e.g. Lamarque et al., 2010; Fiore et al.,
2012; Bowman et al., 2013; Lee et al., 2013; Voulgarakis et al., 2013). Lamarque et al. (2013)
and Young et al. (2013) provided the main characteristics of 16 models and details for the
ACCMIP simulations. Although within the ACCMIP project 16 models are available, due to
the lack of hourly $O_3$ concentration here we only focus on 6 global chemistry models with
different configurations (Table 1).

The length of historical and RCP simulations vary between models, but for all models the
historical runs cover a period centered around 2000, while the time-slice of RCPs is centered
around 2100 (Table 1). As for each model we compare the mean change between the
historical and RCP simulations, a different length in the number of years used in the analysis
does not affect the results.
*Potential ozone injury on vegetation*
The $O_3$ exposure-based index, i.e. AOT40 (ppb h), is a metric used to assess the potential $O_3$
risk to vegetation from local to global scales (Emberson et al., 2014). It is computed as sum





of the hourly exceedances above 40 ppb, for daylight hours (8am-8pm) over species-specific
growing seasons (UNECE, 2010). A recent study over Europe showed how computing
AOT40 only over the growing season (i.e. April-September) would lead to an underestimation
of AOT40 up to 50% for conifer trees, while in case of deciduous trees the underestimation is
much smaller (< 5%, Anav et al., 2016). Besides, it should be noted that in Anav et al. (2016)
the AOT40 is computed year-round when the stomatal conductance is greater than 0. Here,
because of the lack of hourly meteorological data, we can only compute the AOT40 year-
round and during the daylight hours. In case of risk assessment, this approach would lead to a
relevant overestimation of AOT40, mainly over polluted area of NH. Nevertheless, since the
aim of this study is to compare how $O_3$ stress to vegetation changes between historical period
and future, the overestimation of AOT40 does not affect our results. Therefore, we computed
AOT40 as follows:

$$AOT40 = \int_{01jan}^{31dec} \max\big(([O_3] - 40), 0\big) dt \qquad (1)$$

where $[O_3]$ is hourly $O_3$ concentration (ppb) simulated by the models at the lower model layer
and $dt$ is time step (1h). The function "maximum" ensures that only values exceeding 40 ppb
are taken into account. The $O_3$ concentration to be used in AOT40 calculation should be at the
top of the canopy; however, most of models used here provide $O_3$ concentrations at 90-120 m.
Nevertheless, even if the $O_3$ concentration is simulated at different elevations above the sea
level, as for each model we compare the variation between present and future, the change is
consistent because the elevation is the same. For the protection of forests, a critical level of
5,000 ppb.h (or 5 ppm.h) is recommended by UNECE (2010). Within the 2008/50/CE
Directive, the critical level for agricultural crops (3 ppm.h) is adopted as the long-term
objective value for the protection of vegetation by 2020.

From the AOT40, a factor of risk for forests and crops can be computed (Anav et al. 2011;
Proietti et al. 2016). Thus, the potential $O_3$ impact on photosynthetic assimilation ($IO_3$) is
expressed as following:

$IO3 = \alpha \times AOT40$ \qquad (2)

where $\alpha$ is an empirically derived $O_3$ response coefficient representing the proportional
change in photosynthesis per unit of ozone-uptake (Anav et al., 2011). The coefficient for
coniferous trees ($0.7 \times 10^{-6}$ mm$^{-1}$ ppb$^{-1}$) and crops ($3.9 \times 10^{-6}$ mm$^{-1}$ ppb$^{-1}$) are based on the





**Results and Discussion**

Although differences in the simulated global $O_3$ spatial pattern were previously discussed and analyzed (e.g. Lamarque et al., 2013), we show the mean annual $O_3$ concentration at the lower model layer in Figure 1 because $O_3$ concentration explains AOT40 patterns. Then, in Figure 2 we show and discuss the AOT40 spatial and temporal distribution from the ACCMIP models for the historical and RCPs simulations, and finally in Figure 3 we show the percentage of variation of IO3, i.e. the change in the potential impact of $O_3$ on vegetation for the ACCMIP models computed comparing the RCPs simulations with historical runs. A detailed description of each figure, model by model, is included in Supplementary Information (SI).

**Spatial pattern of historical ozone concentration and AOT40**

The highest surface $O_3$ concentrations (Fig. 1) and potential $O_3$ injury (Fig. 2) are found in the NH, highlighting a hemispheric asymmetry. The multi-models $O_3$ mean concentration, averaged over the land points of the domain, is 37.9 ± 4.3 ppb in NH and 22.9 ± 3.8 ppb in SH (Table 3a). The NH extratropics (i.e. mid-latitudes beyond the tropics) has 65% more $O_3$ than the SH extratropics (data not shown). The highest AOT40 values are found in the NH, with an averaged AOT40 of 24.8 ± 10.1 ppm.h in NH and 2.5 ± 1.7 ppm.h in SH (Table 3a).

According to previous studies, the annual mean background $O_3$ concentrations at NH mid-latitudes range between 35 and 50 ppb during the end of the 20[th] century (e.g. Cooper et al., 2012; IPCC, 2014; Lefohn et al. 2014). Similarly, we found historical surface $O_3$ mean concentrations ranging between 35 and 50 ppb and 35-50 ppm.h for AOT40 in the NH, with the highest values occurring over Greenland and in the latitude band 15-45°N, particularly around the Mediterranean basin, Near East, Northern America and over the Tibetan plateau (> 50 ppb and 70 ppm.h) while the lowest $O_3$ burden (15-30 ppb, < 20 ppm.h) was recorded in SH, particularly over Amazon, African and Indonesian rainforests. Tropospheric $O_3$ has a significant source from stratospheric $O_3$ (Parrish et al., 2012) and it can be transported by the large-scale Brewer-Dobson overturning circulation, i.e. an upward motion from the tropics and downward at higher latitudes, resulting in higher $O_3$ concentrations in the extratropics



(Hudson et al., 2006; Seidel et al., 2008; Parrish et al., 2012). The six models are able to
reproduce the spatial pattern of $O_3$ concentration and thus AOT40 worldwide.

The highest historical $O_3$ mean concentrations are observed in GFDL-AM3 and the lowest are
found in MIROC-CHEM. In the early 2000s, the maximum global $O_3$ mean concentration (39
ppb) in GFDL-AM3 is associated to the lowest annual total NOx emissions (46.2 Tg, Table
2a) and low LNOx (4.4 Tg) while the minimum global $O_3$ mean concentration (28 ppb) in
MIROC-CHEM is related to the highest emissions of total NOx per year (57.3 Tg) and
erroneously high LNOx (9.7 Tg per year, Lamarque et al., 2013). MIROC-CHEM simulates
58 gaseous species in the chemical scheme with constant present-day biogenic VOCs
emissions while GFDL-AM3 simulates 81 species (Stevenson et al., 2012; Lamarque et al.,
2013). In GISS-E2-R, the hemispheric asymmetry in $O_3$ is more important with e.g. a mean
concentration of 22 ppb in SH and 42 ppb in NH. A stronger global AOT40 mean (26 ppm.h)
is observed in GISS-E2-R and the lowest (7 ppm.h) in MIROC-CHEM for historical
simulations. Model-to-model differences are observed due to different natural emissions of $O_3$
precursors (e.g. lightning NOx) and the used chemical schemes.

Higher $O_3$ burdens (mean concentration > 50 ppb, AOT40 >70 ppm.h) are simulated at high-
elevation areas, e.g. at Rocky and Appalachian Mountains and over the Tibetan plateau (Fig.
1, Fig. 2).At high-elevation, solar radiation, biogenic VOC emission, exchange between free
troposphere and boundary layer, and stratospheric $O_3$ intrusion within the troposphere are
more important that at the surface layer (Steinbacher et al. 2004; Kulkarni et al., 2011; Lefohn
et al., 2012). Altitude reduces the $O_3$ destruction by deposition and NO (Chevalier et al.,
2007). In addition, due to the high elevation, ambient air remains colder and dryer in summer,
leading to lower summertime $O_3$ losses from photolysis (Helmig et al., 2007). The high-
elevation areas, characterized by higher $O_3$ burdens, are well simulated in GISS-E2-R and
MOCAGE models.

The Tibetan plateau, so-called "ozone valley", is the highest plateau in the world, with a mean
height of 4000 m a.s.l. (Tian et al., 2008) with strong thermal and dynamic influences on
regional and global climate (Chen et al. 2011). High surface $O_3$ mean concentrations (40-60
ppb) were reported in previous studies (e.g. Zhang et al., 2004; Bian et al., 2011; Guo et al.,
2015; Wang et al., 2015). Although this region is remote, road traffic, biofuel energy source,
coalmines and trash burning are prevalent. These pollution sources contribute to significant





amount of NOx, CO and VOCs (Wang et al., 2015). The high $O_3$ levels are attributed to the
combined effects of high-elevation surface, thermal and dynamical forcing of the Tibetan
plateau and *in-situ* photochemical production in the air trapped in the plateau by surrounding
mountains (Guo et al., 2015; Wang et al., 2015). The dynamic effect, associated with the
large-scale circulation, is more important than the chemical effect (Tian et al., 2008; Liu et al.,
2010) and responsible for the high $O_3$ levels over the Tibetan plateau. The six models are able
to well reproduce the high surface $O_3$ mean concentrations (> 50 ppb) over the Tibetan
plateau.

Higher $O_3$ mean concentrations (> 60 ppb) are also observed in Southwestern U.S., at the
stations inland close to Los Angeles, in Northeastern U.S. and East Asia (e.g. Beijing) (Fig.
1). The American Southwest is an $O_3$ precursor hotspot where the industrial sources emit $CH_4$
and VOCs into the air (Jeričević et al., 2013) and the eastern and northern desert areas have
higher ambient $O_3$ than urban areas of southern California due to four factors: on-shore winds,
gasoline reformulation, eastward population expansion and nighttime air chemistry (Arbaugh
and Bytnerowicz, 2003). The surface concentrations show higher $O_3$ levels in areas downwind
of $O_3$ precursor sources, i.e. urban and well-industrialized areas, at distances of hundreds or
even thousands of kilometers due to transport of $O_3$ and precursors, including "reservoir"
species such as PAN, lower $O_3$ titration by NO and higher biogenic VOC emission (Wilson et
al., 2012; Paoletti et al., 2014; Monks et al., 2015; Sicard et al., 2016a).The higher $O_3$ levels
in areas downwind of $O_3$ precursor sources are well simulated in GISS-E2-R and MOCAGE
models.

In the lower troposphere, $O_3$ can be removed by a large number of chemical reactions and by
dry deposition (Sicard et al., 2016c). The $O_3$ dry deposition rates range from 0.01-0.05 cm s$^{-1}$
(oceans and snow) to 0.15-1.80 cm s$^{-1}$ for mixed wood forests (Wesely and Hicks, 2000;
Zhang et al., 2003). The model performance is also related to the parameterization of the dry
deposition rates.

Over Greenland, mean $O_3$ concentrations during the historical runs, ranged from 40 to 55 ppb
(Fig. 1) except in MIROC-CHEM (20-25 ppb). Similarly, Helmig et al. (2007) reported
annual mean of surface $O_3$ concentrations of 47 ppb over Greenland between 2000 and 2005,
particularly at the high-elevation Summit station (3200 m a.s.l.). Several investigations, about
snow photochemical and oxidation processes over Greenland, concluded that photochemical





$O_3$ production can be attributed to high levels of reactive compounds (e.g. oxidized nitrogen
species) present in the surface layer during the sunlit periods due to local sources e.g. NOx
enhancement from snowpack emissions, Peroxyacetyl nitrate (PAN) decomposition, boreal
forest fires or ship emissions (Granier et al., 2006; Stohl et al., 2007; Legrand et al., 2009;
Walker et al., 2012). PAN to NOx ratio increases with increasing altitude and latitude (Singh
et al., 1992). The PAN reservoir for NOx may be responsible for the increase in surface
$O_3$ concentrations at high latitudes (Singh et al., 1992). Local $O_3$ production does not appear to
have an important contribution to the ambient high $O_3$ levels (Helmig et al., 2007), however
the long-range $O_3$ transport can elevate the background concentrations measured at remote
sites, e.g. Greenland (Ellingsen et al., 2008; Derwent et al., 2010). Low dry deposition rates
for $O_3$, the downward transport of stratospheric $O_3$, the photochemical local production and
the large-scale transport (Legrand et al., 2009; Walker et al., 2012; Hess and Zbinden, 2013)
are known factors to explain higher $O_3$ pollution over Greenland.

The surface $O_3$ concentrations (> 40 ppb) and AOT40 (> 60 ppm.h) are higher over deserts,
downwind of $O_3$ precursor sources (e.g. Near East, Sierra Nevada, Colorado Desert), due to
lower $O_3$ dry deposition fluxes, $O_3$ precursors long-range transport from urbanized areas and
high insolation. Around the Mediterranean basin, elevated AOT40 values (> 60 ppm.h) are
recorded, mainly due to the industrial development, road traffic increment, high insolation,
sea/land breeze recirculation and $O_3$ transport (Sicard et al., 2013). All models, except
MIROC-CHEM, are able to well reproduce the high surface $O_3$ mean concentrations over
Greenland and over deserts.

**Projected changes in ozone concentration and AOT40**

Recent studies display a mean global increase in background $O_3$ concentration from a current
level of 35-50 ppb (e.g. IPCC, 2014; Lefohn et al. 2014) to 55-65 ppb (e.g. Wittig et al., 2007)
and up to 85 ppb at NH mid-latitudes by 2100 (IPCC, 2014). During the latter half of the 20[th]
century surface $O_3$ concentrations have increased markedly at NH mid-latitudes (e.g. Oltmans
et al., 2006; Parrish et al., 2012; Paoletti et al., 2014), mainly related to increasing
anthropogenic precursor emissions related to economic growth of industrialized countries
(e.g. Lamarque et al., 2005). Our results indicate that the future projections of the mean
tropospheric $O_3$ concentrations and AOT40 vary considerably with the different scenarios and
models (Fig. 1 and 2). The six models simulate a decrease of $O_3$ concentration by 2100 under
the RCP2.6 and RCP4.5 scenarios, and an increase under the RCP8.5 scenario (Lamarque et





al., 2011). In our study, the averaged relative changes in surface $O_3$ concentration means (and
AOT40) for the different RCPs are: -21% (-75%) for RCP2.6, - 10% (-50%) for RCP4.5 and
+ 14% (+69%) for RCP8.5 with a strong disparity between both hemispheres, e.g. - 8% in SH
and - 25% in NH for RCP2.6 (Tables 3b-c). RCP8.5 is the only scenario to show an increase
in global background $O_3$ levels by 2100 (+ 23% in SH and + 11% in NH).

Under the RCP2.6 scenario, all models predict that tropospheric $O_3$ will strongly decrease
worldwide, except in Equatorial Africa where higher $O_3$ levels are observed in GFDL-AM3,
GISS-E2-R and MOCAGE. In CESM-CAM, GFDL-AM3 and MIROC-CHEM, a
homogeneous decrease in $O_3$ burden is simulated worldwide while in GISS-E2-R, MOCAGE
and UM-CAM, the strongest decrease in surface $O_3$ mean concentrations are found where
high historical $O_3$ concentrations were reported. Under RCP4.5 scenario, the surface $O_3$ mean
concentrations and AOT40 values are lower than historical runs worldwide for all models
except in MOCAGE where deterioration is observed over Canada, Greenland and East Asia.
For all models, the surface $O_3$ levels and AOT40 are higher for RCP8.5 as compared to
historical runs and the highest increases occur in the Northwestern America, Greenland,
Mediterranean basin, Near East and East Asia. The AOT40 values, exceeding 70 ppm.h, are
found over the Tibetan plateau and in Near East and over Greenland. For RCP8.5, GFDL-
AM3 is the most pessimistic model and MIROC-CHEM the most optimistic. By the end of
the 21$^{st}$ century, similar patterns are evident for RCP4.5 compared to RCP2.6 and RCP4.5
simulation is intermediate between RCP2.6 and RCP8.5 ones.

For all models and RCPs, the $O_3$ hot-spots (mean concentrations > 50 ppb and AOT40 > 70
ppm.h) are over Greenland and South Asia, in particular over the Tibetan plateau. The highest
increases are observed in NH, in particular in Northwestern America, Greenland, Near East
and South Asia (> 65 ppb). For the three RCPs, no significant change in tropospheric $O_3$ is
observed in SH and the SH extratropics makes a small contribution to the overall change.

A recent global study showed the geographical patterns of surface air temperature differences
for late 21$^{st}$ century relative to the historical run (1986-2005) in all RCP scenarios (Nazarenko
et al., 2015).The global warming in the RCP2.6 scenario is 2-3 times smaller than RCP4.5
scenario and 4-5 times smaller than RCP8.5 scenario (Nazarenko et al., 2015). For the three
RCPs, the greatest change is observed over the Arctic, above latitude 60°N, and in the latitude
band 15-45°N (IPCC, 2014; Nazarenko et al., 2015). The least warming is simulated over the





large area of the Southern Ocean. For RCP8.5 scenario, the global pattern of surface $O_3$ levels
and AOT40 (Fig. 1-2) is similar to surface air temperature increase distribution. For RCP8.5,
significant increases in air temperature are simulated over latitude 60°N and over the Tibetan
plateau (more than 5°C). An increase of 4-5°C over the Near East, East and South Asia, North
and South Africa and Canada are simulated as well as + 1-3°C for the rest of the world
(Nazarenko et al., 2015). The tropospheric warming is stronger in the latitude band 15-45°N
(Seidel et al., 2008) and Hudson et al. (2006) have demonstrated that $O_3$ trends over a 24-year
period in the NH are due to trends in the relative area of the tropics and mid-latitudes and
Polar Regions.All models are able to reproduce the global pattern of air temperature changes
distribution in agreement with surface $O_3$ concentrations changes.

The spread in precursor emissions (e.g. VOCs, NOx, CO) is due to the range of representation
of biogenic emissions (NOx from soils and lightning, CO from oceans and vegetation) as well
as the complexity of chemical schemes in particular for NMVOCs simulations (e.g. isoprene)
from explicitly specified to fully interactive with climate. RCP2.6 scenario has the lowest $O_3$
precursor concentrations, and RCP8.5 has relatively low NOx, CO and VOCs emissions, but
very high $CH_4$ (Table 2b). The global emissions of NOx (-44%), VOCs (-5%) CO (-40%) and
$CH_4$ burden (-27%) decline, while LNOx increase by e.g. 7% under RCP2.6 (Table 2b). The
CO (-32%) and NOx (-20%) emissions have decreased while LNOX (+33%), VOCS (+1%)
and $CH_4$ burden have increased (+120%) under RCP8.5 scenario (Table 2b). The GISS-E2-R
model shows a greater degree of variation than other models, with a stronger increase in $CH_4$
burden (+ 153%) and in VOCs emissions (+ 20%) for RCP8.5 (Table 2b).

Excluding $CH_4$ burden and VOCs emissions, all the RCPs include reductions and
redistributions of $O_3$ precursor emissions throughout the 21$^{st}$ century, due to the air pollution
control strategies worldwide. The changes in $CH_4$ burden are due to the different climate
policies in model assumptions. In RCP2.6, $CH_4$ emissions decrease steadily throughout the
century, in RCP4.5 it remain steady until 2050 and then decrease (Voulgarakis et al., 2013)
and in RCP8.5 (no climate policy) it rapidly increase compared to 2000. Methane burdens are
fixed in the models with no sources, except for the GISS-E2-R simulations in which surface
$CH_4$ emissions are prescribed for future rather than concentrations (Shindell et al., 2012). The
model chemical schemes vary greatly in their complexity, mainly due to the NMVOCs
simulations (Young et al. 2013). Isoprene dominates the total NMVOCs emissions (Guenther
et al., 1995). Inversely to other models with constant present-day isoprene emissions, the



GISS-ES2-R simulations incorporate climate-driven isoprene emissions, with greater BVOC
emissions by 2100 and a positive change in total VOCs emissions across RCPs, related to the
positive correlation between air temperature and isoprene emission (e.g. Guenther et al., 2006;
Arneth et al., 2011; Young et al., 2013).

For RCP2.6 and RCP4.5 scenarios, there is a widespread decrease in $O_3$ in NH by 2100. The
overall decrease in $O_3$ concentration and AOT40 means for RCP4.5 are about half of that
between RCP2.6 and the historical simulation. For both scenarios, the changes are dominated
by the decrease in $O_3$ precursor emissions in the NH extratropics compared to historical
simulations (Table 2b). In NOx saturated areas, annual mean $O_3$ will slightly increase as a
result of a less efficient titration by NO, but the overall $O_3$ burden will decrease substantially
at hemispheric scale over time (Gao et al., 2013; Querol et al., 2014; Sicard et al., 2016a). In
RCP4.5, Gao et al. (2013) showed that the largest decrease in $O_3$ (4-10 ppb) occurs in summer
at mid-latitudes in the lower troposphere while the $O_3$ concentrations undergo an increase in
winter. During the warm period, the photochemistry plays a major role in the $O_3$ production,
suggesting that the reduction in surface $O_3$ concentrations is in agreement with the large
reduction in anthropogenic $O_3$ precursor emissions (Sicard et al., 2016a) reducing the extent
of regional photochemical $O_3$ formation (e.g. Derwent et al., 2013; Simpson et al., 2014).
Titration effect was also reported by Collette et al. (2012) over Europe by using six chemistry
transport models.

The $O_3$ increase can be also driven by the net impacts of climate change, i.e. increase in
stratospheric $O_3$ intrusion, changing LNOx and impacting reaction rates, through sea surface
temperatures and relative humidity changes (Lau et al., 2006; Voulgarakis et al., 2013; Young
et al., 2013).

Under the RCP8.5 scenario, the increase in surface $O_3$ concentrations, by 14% on average, can
be attributed to the higher $CH_4$ emissions coupled with a strong global warming, exceeding
2°C, and a weakened NO titration by reducing NOx emissions (Stevenson et al., 2013; Young
et al., 2013). The global $CH_4$ burden are 27% and 5% lower than 2000, for the RCP2.6 and
RCP4.5 scenarios respectively while for RCP8.5, the total $CH_4$ burden has more than doubled
compared to early 2000s and LNOx emissions increased by 33% (Table 2b). In addition,
stronger increases are found over the high-elevation Himalayan Plateau reflecting increased
exchange with the free troposphere or stratosphere (Lefohn et al., 2012; Schnell et al., 2016).





Several studies reported an increase in the stratospheric $O_3$ influx and higher stratospheric $O_3$
levels in response to a warming climate (e.g. Hegglin and Shepherd, 2009; Zeng et al., 2010).
The downwards $O_3$ transport from the stratosphere is an important source of tropospheric $O_3$
(Hsu and Prather, 2009; Tang et al., 2011), therefore, stratospheric $O_3$ recovery also plays a
partial role (e.g. + 11% for RCP8.5) in surface $O_3$ burden pattern. As an example, in
MOCAGE, smaller reduction in global $O_3$ mean concentrations (-13%) and higher increase in
stratospheric $O_3$ inputs (+20%) are observed for RCP2.6 (Table 3b). Similarly, for RCP8.5,
the highest increase in $O_3$ mean concentrations (+23%) and stratospheric $O_3$ (+24%) are
recorded in MOCAGE. In addition, lightning NOx emissions show significant upward trend
from 2000 to 2100, in particular for the strongest warming scenario (RPC8.5) with greater
convective and lightning activity (e.g. Williams, 2009; Lamarque et al., 2013). For RCP8.5, a
reduction in surface $O_3$ concentrations is also simulated over the equatorial region, where the
increased relative humidity, in a warmer climate, increases the $O_3$ loss rate (e.g. Johnson et
al., 1999; Zeng and Pyle, 2003).

**Risk areas for vegetation under RCP scenarios**

Figure 3 shows the changes in the potential $O_3$ injury between present and future. It should be
noted that a zero percentage of change (i.e. no change) for IO3, is simulated in sparsely
vegetated regions (e.g. Gobi, Sahara, Near East, Western plateau and Greenland), while the
change can be higher than 100% when the historical $O_3$ concentrations are lower than 40 ppb
(i.e. AOT40 = 0 and IO3 = 0) and the $O_3$ concentrations exceed 40 ppb under RCPs (i.e.
AOT40 > 0, IO3 > 0).

The potential $O_3$ impact for vegetation strongly decreases in NH for RCP2.6, except in
MOCAGE where a slight increase in the risk factor (+ 15 %) is simulated at high latitudes and
in South Asia. Conversely, the areas where the risk for vegetation increases (> 60 %) occur
over Africa (+ 15% to + 60%) for all models, except in CESM-CAM where no change is
observed across Africa. Under RCP4.5 scenario, the strongest increase in potential risk for
vegetation (> + 60 %) is simulated by MOCAGE, markedly different from the other models,
above the latitude 50°N. For all models, the potential $O_3$ impact for vegetation increases
across Africa, from - 15% to + 60% while slight decreases or no change occur worldwide.
Under RCP8.5 scenario, an increase of average $O_3$ over a significant part of the domain is
simulated, therefore the exposure to $O_3$ pollution and impacts on vegetation will increase
worldwide by 2100. An increase of the $O_3$ impacts on vegetation is simulated in Northern



U.S., South America, Asia and Africa while a reduction in particular over Eastern U.S. and
Southeastern China, and a slight increase (+ 15%) or decrease (- 15%) over Europe depending
on the model, are simulated.
In summary, compared to the historical simulations, the averaged relative changes in the $O_3$
risk factor for the different RCPs are: - 61% for RCP2.6, - 47% for RCP4.5 and + 70% for
RCP8.5 (Table 3d). We thus find a significant reduction in risk for vegetation for both
RCP2.6 and RCP4.5 scenarios, except in South Africa and at high-latitudes in MOCAGE
simulations, and a strong increase in global risk under RCP8.5. Under RCP2.6 and RCP4.5
scenarios, IO3 slightly increases in Africa and over North America and Asia (> latitude 60°N)
in MOCAGE. The risk increases over the few areas where the $O_3$ concentrations increased
between the historical period and 2100. Under both scenarios, the strongest reductions in risk
are observed over Amazon, Central Africa and South Asia, i.e. where the $O_3$ concentrations
have strongly declined between historical period and 2100. Under the RCP8.5, the areas
where the highest projected $O_3$ mean concentrations are simulated (e.g. Greenland, deserts)
are not associated to an increase in IO3 due to the absence of vegetation. Under RCP8.5, IO3
increases worldwide while a reduction is simulated over Southeast North America, northern
Amazon, Central Africa and Southeast Asia, and a slighter reduction or a slight increase is
simulated over Western Europe (depending on the model).

The spatial pattern of IO3 is consistent with previous analyses on climate change and $O_3$
impacts on vegetation (e.g. Nemani et al., 2003; Zhu et al., 2016), i.e. the highest reduction in
risk for vegetation, in particular under RCP8.5, occurs over areas where a strong increase in
greening, LAI and NPP is observed due to global change and where a reduction in $O_3$ mean
concentrations is found by 2100 (Fig. 1). The regions with the largest greening trends are in
Southeast North America, northern Amazon, Europe, Central Africa and Southeast Asia with
an average increase of the observed LAI exceeding 0.25 m² m$^{-2}$ per year (Zhu et al., 2016).
The $CO_2$ fertilization effects (70%), nitrogen deposition (9%) and climate change (8%)
explain the observed greening trend (Zhu et al., 2016). The changing climate alone produces
persistent NPP increases and the regions with the highest increase in NPP, ranging from 1.0-
1.5% per year, are in Southeast North America, northern Amazon, Western Europe, Central
Africa and South Asia (Nemani et al., 2003). NPP increased by 6% globally between 1982
and 1999 and the highest increases are observed in tropical regions, with more than 1.5% per
year over Amazon rainforest which accounts for 42% of the global NPP increase (Nemani et



al., 2003). Amazon rainforest is one region where the effects are statistically significant. This
is particularly important owing to the role of the Amazon rainforests in the global carbon
cycle (Zhu et al., 2016). In these areas, the increasing effect of a warming climate on forests
(e.g. increase of greening, LAI) is higher than the reduction in GPP due to $O_3$. Inversely, the
risk for vegetation increases in particular in Africa, e.g. western Africa along the Gulf of
Guinea, in South Brazil and over high-latitudes regions (> 60°N) in North America and Asia
where a reduction or a slight increase in LAI (from - 0.05 to + 0.03 m² m$^{-2}$ per year) and
strong decreases, by 1.0-1.5% per year, in NPP are simulated (Nemani et al., 2003; Zhu et al.,

486    2016).


Our results are not in agreement with the high GPP reduction, due to $O_3$ effects, simulated by
Sitch et al. (2007) between 1901 and 2100, with a projected GPP reduction exceeding 30%
over Western Europe, eastern and western North America, Amazon, central Africa and East
Asia where higher surface $O_3$ mean concentrations were projected. Previous studies reported
that the reductions in GPP simulated by Sitch et al. (2007) are overestimated up to six times
(Ren et al., 2011; Zak et al., 2011; Kvaleveg and Myhre 2013), mainly due to the lack of
empirical data about the response of different species to $O_3$, the fact that a few experiments
have shown no response, e.g. grasslands (Bassin et al., 2013), and the non-inclusion of the
nitrogen limitation of growth (Kvalevag and Myhre, 2013).

The projected land covers widely vary under RCPs (Betts et al., 2015). In RCP2.6 scenario,
the ground surface covered by croplands increases as a result of bio-energy production, with a
more-or-less constant use of grassland. The RCP4.5 scenario focuses on global reforestation
programs as part of global climate policy, as a result, the use of cropland and grassland
decreases. Under RCP8.5, an increase in croplands and grasslands is applied mostly driven by
an increasing global population (van Vuuren et al., 2011). Generally, the risk for vegetation
strongly increases over shrublands (e.g. high-latitude region, Australia, South Africa) and
savannas (e.g. South Brazil, Africa) and the risk decreases over forests, strongly over
evergreen broadleaf forest and deciduous woodland over Africa and Amazon rainforests, and
slighter over needleleaf forests in Northern America (Canada) and Northern Asia. The risk
slightly decreases or slightly increases over grasslands (Central Asia and central Africa and
U.S.). The largest decreases (50-80%) under RCP8.5 occur in Eastern U.S., Europe and
Southeastern China, where the ground is mainly dominated by croplands, in all models except
CESM-CAM.



**Conclusions**


From six global atmospheric chemistry transport models, we illustrate the changes, i.e.
differences for late $21^{st}$ century relative to the historical run, in ground-level $O_3$
concentrations and vegetation impact metric (AOT40). *In fine*, the potential $O_3$ impacts on
vegetation worldwide are investigated to define potential risk areas for vegetation at global
scale by 2100.

The six models are able to well reproduce the spatial pattern of historical $O_3$ concentration
and AOT40 at global scale, in particular GISS-E2-R and MOCAGE are able to simulate the
higher $O_3$ levels in areas downwind of precursor sources and at the high-elevation areas. The
model outputs emphasize the strong asymmetry in the tropospheric $O_3$ distribution between
NH and SH; substantially higher $O_3$ mean concentrations are observed in the NH (ca. 38 ppb),
particularly in the latitude band 15-45°N, than in the SH (ca. 23 ppb). The natural emissions
of $O_3$ precursors (e.g. lightning NOx, CO from oceans, isoprene) as well as the complexity of
chemical schemes are significant sources of model-to-model differences.

In this study, the projected mean tropospheric $O_3$ concentrations and AOT40 dependent on
global and regional emission pathways. Compared to early 2000s, the results suggest changes
in surface $O_3$ of - 9.5 ± 2.0 ppb (NH) and - 1.8 ± 2.1 ppb (SH) in the cleaner RCP2.6 scenario
and of + 4.4 ± 2.8 ppb (NH) and + 5.1 ± 2.1 ppb (SH) in RCP8.5 scenario. For RCP2.6 and
RCP4.5, absolute decreases are observed for the Mediterranean basin and the Western U.S.
due to less precursor emissions in the NH extratropics (e.g. reduction of 5-7 ppb over
Europe). Smaller reduction in surface $O_3$ levels in South and East Asia highlight the smaller
changes in $O_3$ precursor emissions due to the recent emission growth in this region (e.g.
Zhang et al., 2009; Xing et al., 2015). For RCP8.5, all models show climate-driven increases
in ground-level $O_3$ in particular over the Western U.S, Greenland, South Asia and Northeast
China. The changes in surface $O_3$ over North America and Europe ranged from + 1-5 ppb
under RCP8.5. South Asia sees the greatest increase, up to more than 10 ppb for RCP 8.5. The
$O_3$ increase can be attributed to substantial increase in $CH_4$ emissions coupled with a strong
global warming, exceeding 2°C, and a weakened NO titration and a greater stratospheric $O_3$
influx (Kawase et al., 2011; Wild et al., 2012; Young et al., 2013). A decline in $CH_4$
emissions will undoubtedly benefit future $O_3$ control.



The current surface $O_3$ levels (35-50 ppb in NH) are high enough to damage both forests and
crops. About 50% of forests, grasslands and croplands might be exposed to high $O_3$ levels by
the end of the 21$^{st}$ century (Sitch et al., 2007; Wittig et al. 2009). Most important results from
the study are the significant overrun of exposure metric (AOT40) in comparison with the
AOT40-based critical level for the protection of forests (5 ppm.h) and crops (3 ppm.h). The
global models suggest that exposure-based critical levels will be exceeded over many areas of
the NH, and in parts of North America, East and South Asia they may be exceeded by a factor
exceeding 10 under RCP8.5. The critical level were defined for boreal and temperate
deciduous tree species, i.e. more consistent for regions in the latitude band 35-60°N. To
protect vegetation, the current AOT40 index appears inadequate for a realistic quantification
of $O_3$ impacts on vegetation (Paoletti and Manning, 2007; Mills et al., 2011; De Marco et al.,
2015; Sicard et al., 2016b,c). As a result, in the last decade, the United Nations Convention on
Long-Range Transboundary Air Pollution (CLRTAP) has introduced the flux-based metric
for vegetation protection against effects of $O_3$, taking into account the modifying effects of
multiple climatic and phenological factors on $O_3$ uptake (Paoletti and Manning, 2007; Sicard
et al., 2016b,c).
Ozone may be a major threat to biodiversity over large regions of the world (Sicard et al.,
2016b), however the size of these areas remains uncertain. The potential $O_3$ impact on
assimilation, IO3, provides a clear indicator of the potential risk to vegetation. The risk for
vegetation decreases by about 61% and 47% under RCP2.6 and RCP4.5, respectively and
increases by 70% under RCP8.5, compared to early 2000s over the whole domain by 2100
and that the potential risk areas for vegetation vary worldwide according to the dominant
vegetation cover. The strongest increase of the $O_3$ impacts on vegetation is simulated in
Northern America and Asia and central Africa. The highest reduction in risk for vegetation
(i.e. Southeast North America, the northern Amazon, Central Africa and Southeast Asia)
occurs over areas where a strong increase in greening, LAI and NPP is observed and where a
reduction in $O_3$ mean concentrations is found by 2100.

Trees possess a defence capacity, e.g. through antioxidant activity and a capacity of repairing
injured tissues (Paoletti, 2007).The short-term response to $O_3$ is a reduction in productivity of
crops and forests and long-term changes in community composition could be observed due to
species-specific $O_3$-sensitivity (Wittig et al., 2009). Generally, deciduous woodland are highly
$O_3$-sensitive risk areas, grasslands and needleleaf forests are moderately $O_3$-sensitive risk



578 areas while the lower risk areas include evergreen broadleaf forests . However, crops are more

579 sensitive to $O_3$ exposure than trees and deciduous trees are more sensitive than coniferous

580 trees with lower stomatal conductance (Felzer et al., 2004; Ren et al., 2007; Wittig et al. 2009;

581 Anav et al., 2011). To efficiently protect vegetation against $O_3$ pollution, suitable standards

582 taking into account the detoxification processes (e.g. flux-based metric) are urgently needed.

583

584 As the vegetation atmosphere feedbacks are still under investigated, e.g. impacts of changes

585 of vegetation on air chemistry, we recommend the use of improved chemistry-climate

586 modelling system, fully coupled with dynamic vegetation models, to perform high resolution

587 simulations and to better evaluate the regional exposure of ecosystems to air pollution.

588

589 The risk reduction is possible through climate-change mitigation, e.g. reductions in air

590 pollution, and adaptation actions. An efficient reduction in overall $O_3$ levels is expected over

591 North America and Europe in all RCP scenarios and worldwide if $CH_4$ emissions are reduced

592 (e.g. Kirtman et al., 2013; Pfister et al., 2014; Schnell et al., 2016). However, the increasing

593 effect of a warming climate on surface $O_3$ concentrations is higher than the reduction

594 achieved by the decline in $O_3$ precursor emissions (Revell et al., 2015; Hendriks et al., 2016),

595 therefore, climate change and the measures and policies in e.g. Asia will need to be factored

596 into future $O_3$ policies (Wilson et al., 2012; Lefohn and Cooper, 2015). Many ecosystems

597 worldwide are unprotected from $O_3$ due to the lack of international efforts (Emberson et al.,

598 2014). To be efficient, the mitigation actions for $O_3$ impacts on biodiversity must be as part of

599 international emission reduction programmes.

600

601 **Acknowledgements**

602 This work was carried out with the contribution of the LIFE financial instrument of the

603 European Union (LIFE15 ENV/IT/183) in the framework of the MOTTLES project

604 "Monitoring ozone injury for setting new critical levels" and published within the

605 International Union of Forest Research Organizations (IUFRO) Task Force on Climate

606 Change and Forest Health.




**Bibliographic references**
**Ainsworth E.A**., Yendrek C.R., Sitch S., Collins, W.J., Emberson L.D., 2012, "The effect of
Tropospheric Ozone on Net Primary Productivity and Implications for Climate Change".
Annu. Rev. Plant Biol. 63: 637-661
**Anav A**., Menut L., Khvorostyanov D., Viovy N., 2011, "Impact of tropospheric ozone on the
Euro-Mediterranean vegetation". Global Change Biol. 17: 2342-2359
**ArbaughM.J**., and Bytnerowicz A., 2003, "Ambient ozone patterns and effects over the
Sierra Nevada: synthesis and implications for future research". In: A. Bytnerowicz, M.
Arbaugh, R. Alonso (eds), Ozone Air Pollution in the Sierra Nevada: Distribution and Effects
on Forests, Developments in Environmental Science, vol. 2, Elsevier, Amsterdam, 249-261
**Arneth A**., Schurgers G., Lathière J., Duhl T., Beerling D. J., et al., 2011, "Global terrestrial
isoprene emission models: sensitivity to variability in climate and vegetation". Atmos. Chem.
Phys. 11: 8037-8052
**Arneth A**., Schurgers G., Hickler T., Miller P.A., 2008, "Effects of species composition, land
surface cover, $CO_2$ concentration and climate on isoprene emissions from European forests".
Plant Biol. 10: 150-162
**Ashmore M.R**., 2005, "Assessing the future global impacts of ozone on vegetation". Plant
Cell Environ. 28: 949-964
**Ashworth K**., Wild O., Hewitt C.N., 2013, "Impacts of biofuel cultivation on mortality and
crop yields". Nat. Clim. Change 3: 492-496
**Bassin S**., Volk M., Fuhrer J., 2013, "Species composition of subalpine grassland is sensitive
to nitrogen deposition, but not ozone, after seven years of treatment".  Ecosystems 16: 1105-
630 1117
**Betts R.A**., Golding N., Gonzalez P., Gornall J., Kahana R., et al., 2015, "Climate and land
use change impacts on global terrestrial ecosystems and river flows in the HadGEM2-ES
Earth system model using the representative concentration pathways". Biogeosciences 12:
634 1317-1338
**Bian J**., Yan R., Chen H., Lü D., Massie S.T., 2011, "Formation of the summertime ozone
valley over the Tibetan Plateau: The Asian summer monsoon and air column variations".
Adv. Atmos. Sci. 28: 1318-1325
**Bowman K.W**., Shindell D.T., Worden H.M., Lamarque J.F., Young P.J., 2013, "Evaluation
of ACCMIP outgoing longwave radiation from tropospheric ozone using TES satellite
observations". Atmos. Chem. Phys. 13: 4057-4072
**Clifton O.E**., Fiore A.M., Correa G., Horowitz L.W., Naik V., 2014, "Twenty-first century
reversal of the surface ozone seasonal cycle over the northeastern United States". Geophys.
Res. Lett. 41: 7343-7350
**Chen X.L**., Ma Y.M., Kelder H., Su Z., Yang K., 2011, "On the behaviour of the tropopause
folding events over the Tibetan Plateau". Atmos. Chem. Phys. 11: 5113-5122
**Chevalier A**., Gheusi F., Delmas R., Ordóñez C., Sarrat C., et al., 2007, "Influence of altitude
on ozone levels and variability in the lower troposphere: a ground-based study for Western
Europe over the period 2001-2004". Atmos. Chem. Phys. 7: 4311-4326
**Colette A**., Granier C., Hodnebrog Ø., Jakobs H., Maurizi A., et al., 2012, "Future air quality
in Europe: a multi-model assessment of projected exposure to ozone". Atmos. Chem. Phys.
651 12: 10613-10630
**Cooper O.R**., Parrish D.D., Ziemke J., Balashov N.V., Cupeiro M., 2014, "Global
distribution and trends of tropospheric ozone: An observation-based review". Elementa:
Science of the Anthropocene 2: 000029
**Cooper O.R**., Sweeney C., Gao R.S., Tarasick D., Leblanc T., 2012, "Long-term ozone
trends at rural ozone monitoring sites across the United States, 1990-2010". J. Geophys. Res.
117: D22307





**Cubasch** U., Wuebbles D., Chen D., Facchini M.C., Frame D., et al., 2013, "Introduction, in
Climate Change 2013: The Physical Science Basis". Contribution of Working Group I to the
Fifth Assessment Report of the Intergovernmental Panel on Climate Change, edited by T. F.
Stocker et al., Cambridge Univ. Press, Cambridge, U. K. and New York
**De Marco A**., Sicard P., Vitale M., Carriero G., Renou C., et al., 2015, "Metrics of ozone risk
assessment for Southern European forests: canopy moisture content as a potential plant
response indicator". Atmos. Environ. 120: 182-190
**Derwent R.G**., Utembe S.R., Jenkin M.E., Shallcross D.E., 2015, "Tropospheric ozone
production regions and the intercontinental origins of surface ozone over Europe".Atmos.
Environ. 112: 216-224
**Derwent R.G**., Manning A.J., Simmonds P.G., Spain T.G., O'Doherty S., 2013, "Analysis
and interpretation of 25 years of ozone observations at the Mace Head Atmospheric Research
Station on the Atlantic Ocean coast of Ireland from 1987 to 2012". Atmos. Environ. 80: 361-
671 368

**Derwent R.G**., Witham C.S., Utembe S.R., Jenkin M.E., Passant N.R., 2010, "Ozone in
Central England: the impact of 20 years of precursor emission controls in Europe". Environ.
Sci. Policy 13: 195-204
**Donner L.J**., Wyman B.L., Hemler R.S., Horowitz L.W., Ming Y., et al., 2011, "The
dynamical core, physical parameterizations, and basic simulation characteristics of the
atmospheric component AM3 of the GFDL Global Coupled Model CM3". J. Climate 24:
678 3484-3519

**European Environment Agency**, 2015 "Air quality in Europe - 2015 report".ISBN 978-92-
9213-702-1. Report No 5/2015
**Ellingsen K**., Gauss M., Van Dingenen R., Dentener F.J., Emberson L., et al., 2008, "Global
ozone and air quality: a multi-model assessment of risks to human health and crops". Atmos.
Chem. Phys. 8: 2163-2223
**Emberson L.D.**, Fuhrer J., Ainsworth L., Ashmore M.R., 2014, "Biodiversity and Ground-
level Ozone". Report UNEP/CBD/SBSTTA/18/INF/17. Convention on Biological Diversity,
18th meeting, Montreal, 23-28 June 2014
**Fares S**., Vargas R., Detto M., Goldstein A.H., Karlik J., et al., 2013, "Tropospheric ozone
reduces carbon assimilation in trees: estimates from analysis of continuous flux
measurements". Global Change Biol.19: 2427-2443
**Federal Register**, 2015, "National Ambient Air Quality Standards for Ozone". 40 CFR Part
50, 51, 52, 53, and 58, pp 65292-65468
**Felzer B.S.F**., Kicklighter D.W., Melillo J.M., Wang C., Zhuan Q., et al., 2004, "Ozone
effects on net primary production and carbon sequestration in the conterminous United States
using a biogeochemistry model". Tellus B 56: 230-248
**Fiore A.M**., Naik V., Leibensperger E.M., 2015, "Air quality and climate connections". J. Air
Waste Manage. Assoc. 65: 645-685
**Fiore A.M.,** Naik V.,Spracklen D.V., Steiner A., Unger N. et al., 2012, "Global air quality
and climate". Chem. Soc. Rev. 41: 6663-6683
**Fiscus E.L.**, Booker F.L., Burkey K.O., 2005, "Crop responses to ozone: uptake, modes of
action, carbon assimilation and partitioning". Plant Cell Environ. 28: 997-1011
**Gao Y**., Fu J.S., Drake J.B., Lamarque J.F., Liu Y., 2013, "The impact of emission and
climate change on ozone in the United States under representative concentration pathways
(RCPs)". Atmos. Chem. Phys. 13: 9607-9621
**Granier C**., Niemeier U., Jungclaus J.H., Emmons L., Hess P., et al., 2006, "Ozone pollution
from future ship traffic in the Arctic northern passages". Geophys. Res. Lett. 33, doi:
10.1029/2006GL026180



**Guenther A.B**., Karl T., Harley P., Wiedinmyer C., Palmer P.I., Geron C., 2006, "Estimates of global terrestrial isoprene emissions using MEGAN (Model of Emissions of Gases and Aerosols from Nature)". Atmos. Chem. Phys. 6: 3181-3210

**Guenther A.B**., Hewitt C.N., Erickson D., Fall R., Geron, C., et al., 1995, "A global model of natural volatile organic compound emissions". J. Geophys. Res. 100: 8873-8892

**Guo D**., Su Y., Shi C., Xunn J., Powell Jr. A.M., 2015, "Double core of ozone valley over the Tibetan Plateau and its possible mechanisms". Journal of Atmospheric and Solar-Terrestrial Physics 130: 127-131

**Hegglin M.I**. and Shepherd T.G., 2009, "Large climate-induced changes in ultraviolet index and stratosphere-to-troposphere ozone flux". Nature Geosci. 2: 687

**Helmig D**., Oltmans S.J., Morse T.O., Dibb J.E., 2007, "What is causing high ozone at Summit, Greenland?". Atmos. Environ. 41: 5031-5043

**Hendriks C.**, Forsell N., Kiesewetter G., Schaap M., Schöpp W., 2016, "Ozone concentrations and damage for realistic future European climate and air quality scenarios". Atmos. Environ. 144: 208-219

**Hess P.G**. and Zbinden R., 2013, "Stratospheric impact on tropospheric ozone variability and trends: 1990-2009". Atmos. Chem. Phys. 13: 649-674

**Holland M**., Kinghorn S., Emberson L., Cinderby S., Ashmore M., et al., 2006, "Development of a framework for probabilistic assessment of the economic losses caused by ozone damage to crops in Europe". UNECE International Cooperative Programme on Vegetation. Contract Report EPG 1/3/205. CEH Project No: C02309NEW

**Hsu J**. and Prather M.J., 2009, "Stratospheric variability and tropospheric ozone". J. Geophys. Res. 114: D06102

**Hu X.M**, Klein Petra M., Xue M. et al., 2013, "Impact of the vertical mixing induced by low-level jets on boundary layer ozone concentration". Atmos. Environ. 70: 123-130

**Hudson R.D.,** Andrade M.F., Follette M.B., Frolov A.D., 2006, "The total ozone field separated into meteorological regimes – Part II: Northern Hemisphere mid-latitude total ozone trends". Atmos. Chem. Phys. 6: 5183-5191

**IPCC**, Intergovernmental Panel on Climate Change, 2014, "Summary for Policymakers". In: "Climate Change 2014: Impacts, Adaptation and Vulnerability". Contribution of Working Group II to the Fifth Assessment Report of the Intergovernmental Panel on Climate Change. Cambridge University Press, Cambridge, UK

**Jeričević A**., Koračin D., Jiang J., Chow J., Watson J., et al., 2013, "Air Quality Study of High Ozone Levels in South California". Part of the series NATO Science for Peace and Security Series C: Environmental Security. Air Pollution Modeling and its Application XXII: 629-633

**Johnson C.E**., Collins W.J., Stevenson D.S., Derwent R.G., 1999, "Relative roles of climate and emissions changes on future tropospheric oxidant concentrations". J. Geophys. Res. 104: 18631-18645

**Josse B**., Simon P., Peuch V.H., 2004, "Radon global simulations with the multiscale chemistry and transport model MOCAGE". Tellus-B 56: 339-356

**Kawase H**., Nagashima T., Sudo K., Nozawa T., 2011, "Future changes in tropospheric ozone under Representative Concentration Pathways (RCPs)". Geophys. Res. Lett. 38: L05801

**Kelly J**., Makar P.A., Plummer D.A., 2012, "Projections of mid-century summer air-quality for North America: effects of changes in climate and precursor emissions". Atmos. Chem. Phys. 12: 5367-5390

**Kirtman B**., Power S.B., Adedoyin J.A., Boer G.J., Bojariu R., et al., 2013, "Near-term climate change: Projections and predictability, in Climate Change 2013: The Physical Science Basis". Contribution of Working Group I to the Fifth Assessment Report of the



Intergovernmental Panel on Climate Change, edited by T.F. Stocker et al., Cambridge Univ. Press, Cambridge, U. K., and New York

**Klingberg J**., Engardt M., Karlsson P.E., Langner J., Pleijel H., 2014, "Declining ozone exposure of European vegetation under climate change and reduced precursor emissions". Biogeosciences 11: 5269-5283

**Krinner G**., Viovy N., de Noblet-Ducoudré N., Ogée J., Polcher J., et al., 2005, "A dynamic global vegetation model for studies of the coupled atmosphere-biosphere system". Global Biogeochem. Cy. 19: GB1015

**Kulkarni P.S**., Bortoli D., Domingues A., Silva A.M., 2015, "Surface Ozone Variability and Trend over Urban and Suburban Sites in Portugal". Aerosol Air Qual. Res.: 1-15

**Kulkarni P.S**., Bortoli D., Salgado R., Anton M., Costa M.J., et al., 2011, "Tropospheric ozone variability over the Iberian Peninsula". Atmos. Environ. 45: 174-182

**Kvalevag M.M**. and Myrhe G., 2013, "The effect of carbon-nitrogen coupling on the reduced land carbon sink caused by ozone". Geophys. Res. Lett. 40: 3227-3231

**Lamarque J.F**., Shindell D.T., Josse B., Young P.J., Cionni I., et al., 2013, "The Atmospheric Chemistry and Climate Model Intercomparison Project (ACCMIP): overview and description of models, simulations and climate diagnostics". Geosci. Model Dev. 6: 179-206

**Lamarque J.F.**, Emmons L.K., Hess P.G., Kinnison D.E., Tilmes, S., et al., 2012, "CAM-chem: description and evaluation of interactive atmospheric chemistry in the Community Earth System Model". Geosci. Model Dev. 5: 369-411

**Lamarque J.F**., Bond T.C., Eyring V., Granier C., Heil A., et al., 2010, "Historical (1850–2000) gridded anthropogenic and biomass burning emissions of reactive gases and aerosols: methodology and application". Atmos. Chem. Phys. 10: 7017-7039

**Lamarque J.F**., Hess P.G., Emmons L.K., Buja L.E., Washington W.M., Granier C., 2005, "Tropospheric ozone evolution between 1890 and 1990". J. Geophys. Res. 110: D08304

**Langner J**., Engardt M., Baklanov A., Christensen J.H., Gauss M., et al., 2012, "A multi-model study of impacts of climate change on surface ozone in Europe". Atmos. Chem. Phys. 12: 10423-10440

**Lau N.C**., Leetmaa A., Nath M.J., 2006, "Attribution of atmospheric variations in the 1997-2003 period to SST anomalies in the Pacific and Indian Ocean basins". J. Climate 19: 3607-3628

**Lee Y.H**.and Adams P.J., 2011, "A fast and efficient version of the two-moment aerosol sectional (TOMAS) global aerosol microphysics model". Aerosol Sci. Tech. 46: 678-689

**Lefohn A.S**., Malley C.S., Simon H., Wells B., Xu X., et al., 2017, "Responses of human health and vegetation exposure metrics to changes in ozone concentration distributions in the European Union, United States, and China". Atmos. Environ. 152: 123-145

**Lefohn A.S.** and Cooper O.R, 2015, "Introduction to the Special Issue on Observations and Source Attribution of Ozone in Rural Regions of the Western United States". Atmos. Environ. 109: 279-281.

**Lefohn A.S**., Emery C., Shadwick D., Wernli H., Jung J., OltmansS.J.,2014, "Estimates of background surface ozone concentrations in the United States based on model-derived source apportionment". Atmos. Environ. 84:275-288.

**Lefohn A.S**., Wernli H., Shadwick D., Oltmans S.J., Shapiro M., 2012, "Quantifying the frequency of stratospheric-tropospheric transport affecting enhanced surface ozone concentrations at high- and low-elevation monitoring sites in the United States". Atmos. Environ. 62: 646-656

**Lefohn A.S**., Shadwick D., Oltmans S.J., 2010, "Characterizing changes in surface ozone levels in metropolitan and rural areas in the United States for 1980-2008 and 1994-2008". Atmos. Environ. 44: 5199-5210



**Legrand M.**,Preunkert S., Jourdain B., Gallée H., Goutail F., et al., 2009, "Year-round record
of surface ozone at coastal (Dumont d'Urville) and inland (Concordia) sites in East
Antarctica". J. Geophys. Res. 114: doi: 10.1029/2008JD011667
**Liu C.**, Liu Y., Cai Z., Gao S., Bian J., et al., 2010, "Dynamic formation of extreme ozone
minimum events over the Tibetan Plateau during northern winters 1987-2001". J. Geophys.
Res. 115: D18311
**Meinshausen M.**, Wigley T.M.L., Raper S.C.B., 2011, "Emulating atmosphere-ocean and
carbon cycle models with a simpler model, MAGICC6 - Part 2: Applications". Atmos. Chem.
Phys. 11: 1457-1471
**Mills G.**, Hayes F., Simpson D., Emberson L., Norris D., et al., 2011, "Evidence of
widespread effects of ozone on crops and (semi-)natural vegetation in Europe (1990-2006) in
relation to AOT40 and flux-based risk maps". Global Change Biol. 17: 592-613
**Monks P.S**., Archibald A.T., Colette A., Cooper O., Coyle M., et al., 2015, "Tropospheric
ozone and its precursors from the urban to the global scale from air quality to short-lived
climate forcer". Atmos. Chem. Phys. 15: 8889-8973
**Myhre G**., Shindell D., Bréon F.M., Collins W., Fuglestvedt J., et al., 2013, "Anthropogenic
and Natural Radiative Forcing". In: Climate Change 2013: The Physical Science Basis.
Contribution of Working Group I to the Fifth Assessment Report of the Intergovernmental
Panel on Climate Change. Cambridge University Press, Cambridge, United Kingdom and
New York, USA
**Naik V**., Voulgarakis A., Fiore A.M., Horowitz L.W., Lamarque J.F., et al., 2012,
"Preindustrial to present day changes in tropospheric hydroxyl radical and methane lifetime
from the Atmospheric Chemistry and Climate Model Intercomparison Project (ACCMIP)".
Atmos. Chem. Phys. Discuss. 12: 30755-30804
**Nazarenko L**., Schmidt G.A., Miller R.L., Tausnev N., Kelley M., et al., 2015, "Future
climate change under RCP emission scenarios with GISS ModelE2". J. Adv. Model. Earth
Syst. 7: 244-267
**Nemani R.R**., Keeling C.D., Hashimoto H., Jolly W.M., Piper S.C., et al., 2003, "Climate-
Driven Increases in Global Terrestrial Net Primary Production from 1982 to 1999".Science
836 300: 1560-1563
**Ollinger S.V**., Aber J.D., Reich P.B., 1997, "Simulating ozone effects on forest productivity:
interactions among leaf, canopy, and stand-level processes". Ecol. Appl. 7: 1237-1251.
**Oltmans S.J**., Lefohn A.S., Harris J.M., Galbally I., Scheel H.E., et al., 2006, "Long-term
changes in tropospheric ozone". Atmos. Environ. 40: 3156-3173
**Paoletti E**., De Marco A., Beddows D.C.S., Harrison R.M., Manning W.J., 2014, "Ozone
levels in European and USA cities are increasing more than at rural sites, while peak values
are decreasing". Environ. Pollut. 192: 295-299
**Paoletti E.**, Contran N., Bernasconi P., Günthardt-Goerg M.S., Vollenweider P., 2009,
"Structural and physiological responses to ozone in Manna ash (*Fraxinus ornus L.*) leaves in
seedlings and mature trees under controlled and ambient conditions". Sci. Total Environ. 407:
847 1631-1643
**Paoletti E**. and Manning W.J., 2007, "Toward a biologically significant and usable standard
for ozone that will also protect plants". Environ. Pollut. 150: 85-95
**Paoletti E**., 2006, "Impact of ozone on Mediterranean forest: A review". Environ. Pollut. 144:
851 463-474
**Parrish D.D**., Law K.S., Staehelin J., Derwent R., Cooper O.R., et al., 2012, "Long-term
changes in lower tropospheric baseline ozone concentrations at northern mid-latitudes".
Atmos. Chem. Phys. 12: 11485-11504
**Pfister G.G**., Walters S., Lamarque J.F., Fast J., Barth M.C., et al., 2014, "Projections of
future summertime ozone over the U.S". J. Geophys. Res. Atmos. 119: 5559-5582



**Price C**. and Rind D.H., 1992, "A simple lightning parameterization for calculating global
lightning distributions". J. Geophys. Res., 97: 9919-9933
**Proietti C**., Anav A., De Marco A., Sicard P., Vitale M., 2016, "A multi-sites analysis on the
ozone effects on Gross Primary Production of European forests". Sci. Total Environ. 556: 1-
861 11
**Querol X**., Alastuey A., Pandolfi M., Reche C., Pérez N., et al., 2014, "2001-2012 trends on
air quality in Spain". Sci. Total Environ. 490: 957-969.
**Reich P.B**., 1987, "Quantifying plant response to ozone: a unifying theory". Tree Physiol. 3:
865 63-91
**Ren W**., Tian H., Liu M., Zhang C., Chen G., et al., 2007, "Effects of tropospheric ozone
pollution on net primary productivity and carbon storage in terrestrial ecosystems of China".
J. Geophys. Res. 112: 1-17
**Revell L.E**., Tummon F., Stenke A., Sukhodolov T., Coulon A., et al., 2015, "Drivers of the
tropospheric ozone budget throughout the 21$^{st}$century under the medium-high climate
scenario RCP 6.0". Atmos. Chem. Phys. 15: 5887-5902
**Riahi K**., Rao S., Krey V., Cho C., Chirkov V., et al., 2011, "RCP 8.5 - A scenario of
comparatively high greenhouse gas emissions". Climatic Change 109: 33-57
**Rieder H.E**., Fiore A.M., Horowitz L.W., Naik V., 2015, "Projecting policy-relevant metrics
for high summertime ozone pollution events over the eastern United States due to climate and
emission changes during the 21$^{st}$century". J. Geophys. Res. Atmos. 120: 784-800
**Ridley B.A**., Pickering K.E., Dye, J.E., 2005, "Comments on the parameterization of
lightning-produced NO in global chemistry-transport models". Atmos. Environ. 39: 6184-
879 6187
**Sanderson M.G**., Collins W.J., Hemming D.L., Betts R.A., 2007, "Stomatal conductance
changes due to increasing carbon dioxide levels: Projected impact on surface ozone levels".
Tellus 59B: 404-411
**Sanderson M.G**., Jones C.D., Collins W.J., Johnson C.E., Derwent R.G., 2003, "Effect of
climate change on isoprene emissions and surface ozone levels". Geophys. Res. Lett. 30: 1936
**Schnell J.L**., Prather M.J., Josse B., Naik V., Horowitz L.W., et al., 2016, "Effect of climate
change on surface ozone over North America, Europe, and East Asia". Geophys. Res. Lett.43:
L068060
**Seidel D.J**., Fu Q., Randel W.J., Reichler T.J., 2008, "Widening of the tropical belt in a
changing climate". Nat. Geosci 1: 21-4
**Shindell D.T**., Lamarque J.F., Schulz M., Flanner M., Jiao C., et al., 2012, "Radiative forcing
in the ACCMIP historical and future climate simulations".Atmos. Chem. Phys. Discuss. 12:
892 21105-21210
**Shindell D.T**., Faluvegi G., Stevenson D.S., Krol M.C., Emmons L.K., et al., 2006, "Multi-
model simulations of carbon monoxide: Comparison with observations and projected near-
future changes". J. Geophys. Res. 111: D19306
**Sicard P**., Serra R., Rossello P., 2016a, "Spatiotemporal trends of surface ozone
concentrations and metrics in France". Environ. Res. 149: 122-144
**Sicard P**, Augustaitis A., Belyazid S., Calfapietra C., De Marco A., et al., 2016b, "Global
topics and novel approaches in the study of air pollution, climate change and forest
ecosystems". Environ. Pollut.213: 977-987
**Sicard P**., De Marco A., Dalstein-Richier L., Tagliaferro F., Paoletti E., 2016c, "An
epidemiological assessment of stomatal ozone flux-based critical levels for visible ozone
injury in Southern European forests". Sci. Total Environ. 541: 729-741
**Sicard P**., De Marco A., Troussier F., Renou C., Vas N., Paoletti E., 2013, "Decrease in
surface ozone concentrations at Mediterranean remote sites and increase in the cities". Atmos.
Environ. 79: 705-715



**Sicard P.**, Vas N., Dalstein-Richier L., 2011, "Annual and seasonal trends for ambient ozone concentration and its Impact on Forest Vegetation in Mercantour National Park (South-eastern France) over the 2000-2008 period". Environ. Pollut. 159: 351-362

**Sicard P.**, Coddeville P., Galloo J.C., 2009, "Near-surface ozone levels and trends at rural stations in France over the 1995-2003 period". Environ. Monit. Assess. 156: 141-157

**Simpson D.**, Arneth A., Mills G., Solberg S., Uddling J., 2014, "Ozone - the persistent menace: interactions with the N cycle and climate change". Curr. Opin. Env. Sust. 9-10: 9-19

**Singh H.B.**, Herlth D., O'Hara D., Zahnle K., Bradshaw J.D., et al., 1992, "Relationship of Peroxyacetyl nitrate to active and total odd nitrogen at northern high latitudes: influence of reservoir species on NOx and $O_3$". J. Geophys. Res. 97:16523-30

**Sitch** S., Cox P.M., Collins W.J., Huntingford C., 2007, "Indirect radiative forcing of climate change through ozone effects on the land-carbon sink". Nature 448: 791-794

**Steinbacher M.**, Henne S., Dommen J., Wiesen P., Prevot A.S.H., 2004, "Nocturnal trans-alpine transport of ozone and its effects on air quality on the Swiss Plateau". Atmos. Environ. 38: 4539-4550

**Stevenson D.S.**, Young P.J., Naik V., Lamarque J.F., Shindell D.T., et al., 2013, "Tropospheric ozone changes, radiative forcing and attribution to emissions in the Atmospheric Chemistry and Climate Model Inter-comparison Project (ACCMIP)". Atmos. Chem. Phys. 13: 3063-3085

**Stevenson D.S.**, Young P.J., Naik V., Lamarque J.F., Shindell D.T., et al., 2012, "Tropospheric ozone changes, radiative forcing and attribution to emissions in the Atmospheric Chemistry and Climate Model Inter-comparison Project (ACCMIP)". Atmos. Chem. Phys. Discuss. 12: 26047-26097

**Stevenson D.S.**, Dentener F.J., Schultz M.G., Ellingsen K., van Noije T.P.C., et al., 2006, "Multi-model ensemble simulations of present-day and near-future tropospheric ozone". J. Geophys. Res. 111: D08301

**Stevenson D.S.**, Johnson C.E., Collins W.J., Derwent R.G., Edwards J.M., 2000, "Future estimates of tropospheric ozone radiative forcing and methane turnover – The impact of climate change". Geophys. Res. Lett. 27: 2073-2076

**Stohl A.**, Berg T., Burkhart J.F., Fjaeraa A.M., Forster C., et al., 2007, « Arctic smoke - record high air pollution levels in the European Arctic due to agricultural fires in Eastern Europe in spring 2006". Atmos. Chem. Phys. 7: 511-534

**Tang Q.**, Prather M.J., Hsu J., 2011, "Stratosphere-troposphere exchange ozone flux related to deep convection". Geophys. Res. Lett. 38: L03806

**Teyssèdre H.**, Michou M., Clark H.L., Josse B., Karcher F., et al., 2007, "A new tropospheric and stratospheric Chemistry and Transport Model MOCAGE-Climat for multi-year studies: evaluation of the present-day climatology and sensitivity to surface processes". Atmos. Chem. Phys. 7: 5815-5860

**Tian W.**, Chipperfield M., Huang Q., 2008, "Effects of the Tibetan Plateau on total column ozone distribution". Tellus 60B: 622-635

**UNECE**, United Nations Economic Commission for Europe. Convention on Long-Range Trans-boundary Air Pollution, 2010, "Mapping Critical Levels for Vegetation". nternational Cooperative Programme on Effects of Air Pollution on Natural Vegetation and Crops, Bangor, UK

**van Vuuren D.**, Edmonds J., Kainuma M., Riahi K., Thomson A., et al., 2011, "The representative concentration pathways: an overview". Climatic Change 109: 5-31

**Voulgarakis A.**, Naik V., Lamarque J.F., Shindell D.T., Young P.J. et al., 2013, "Analysis of present day and future OH and methane lifetime in the ACCMIP simulations". Atmos. Chem. Phys. 13: 2563-2587





**Walker T.W**.,Jones D.B.A., Parrington M., Henze D.K., Murray L.T., et al., 2012, "Impacts of mid-latitude precursor emissions and local photochemistry on ozone abundances in the Arctic". Journal of Geophysical Research: Atmospheres 117, doi: 10.1029/2011JD016370

**Wang Q.Y**., Gao R.S., Cao J.J., Schwarz J.P., Fahey D.W., et al. 2015, "Observations of high level of ozone at Qinghai Lake basin in the northeastern Qinghai-Tibetan Plateau, western China". J. Atm. Chem. 72: 19-26

**Wang X**.and Mauzerall D.L., 2004, "Characterizing distributions of surface ozone and its impact on grain production in China, Japan and South Korea: 1900 and 2020". Atmos. Environ. 38: 4383-4402

**Watanabe S**., Hajima T., Sudo K., Nagashima T., Takemura T., et al., 2011, "MIROC-ESM 2010: model description and basic results of CMIP5-20c3m experiments". Geosci. Model Dev. 4: 845-872

**Wesely M.L**. and Hicks B.B., 2000, "A review of the current status of knowledge in dry deposition". Atmos. Environ. 34: 2261-2282

**Wild O**., Fiore A.M., Shindell D.T., Doherty R.M., Collins W.J., et al., 2012, "Modelling future changes in surface ozone: a parameterized approach". Atmos. Chem. Phys. 12: 2037-2054

**Wild O**., 2007, "Modelling the global tropospheric ozone budget: exploring the variability in current models". Atmos. Chem. Phys. 7: 2643-2660

**Williams E.R**., 2009, "The global electrical circuit: A review". Atmos. Res., 91: 140-152.

**Wilson R.C**., Fleming Z. L., Monks P. S., Clain G., Henne S., et al., 2012, "Have primary emission reduction measures reduced ozone across Europe? An analysis of European rural background ozone trends 1996-2005".Atmos. Chem. Phys. 12: 437-454

**Wittig V.E.**, Ainsworth E.A., Naidu S.L., Karnosky D.F., Long S.P., 2009, "Quantifying the impact of current and future tropospheric ozone on tree biomass, growth, physiology and biochemistry: a quantitative meta-analysis". Global Change Biol.15: 396-424

**Wittig** V.E., Ainsworth E.A., Long S.P., 2007,"To what extent do current and projected increases in surface ozone affect photosynthesis and stomatal conductance of trees? A meta-analytic review of the last 3 decades of experiments". Plant, Cell Environ. 30: 1150-1162

**Xing J**., Mathur R., Pleim J., C. Hogrefe, Gan C.M., et al., 2015, "Observations and modeling of air quality trends over 1990–2010 across the Northern Hemisphere: China, the United States and Europe". Atmos. Chem. Phys. 15: 2723-2747

**Young P.J**., Archibald A.T., Bowman K.W., Lamarque J.F., Naik V., et al., 2013, "Preindustrial to end 21st century projections of tropospheric ozone from the Atmospheric Chemistry and Climate Model Intercomparison Project (ACCMIP)". Atmos. Chem. Phys. 13: 2063-2090

**Zak D.R**., Pregitzer K.S., Kubiske M.E., Burton A.J., 2011, "Forest productivity under elevated $CO_2$ and $O_3$; positive feedbacks to soil N cycling sustain decade-long net primary productivity enhancement by $CO_2$. Ecology Letters 14: 1220-1226

**Zeng G**., Morgenstern O., Braesicke P., Pyle J.A., 2010, "Impact of stratospheric ozone recovery on tropospheric ozone and its budget". Geophys. Res. Lett. 37: L09805

**Zeng G**., Pyle J.A., Young P. J., 2008, "Impact of climate change on tropospheric ozone and its global budgets, Atmos. Chem. Phys. 8: 369-387

**Zeng G**. and Pyle J.A., 2003, "Changes in tropospheric ozone between 2000 and 2100 modeled in a chemistry-climate model". Geophys. Res. Lett. 30: 1392

**Zhang Q.**, Streets D.G., Carmichael G.R., He K.B., Huo H., et al., 2009, "Asian emissions in 2006 for the NASA INTEX-B mission". Atmos. Chem. Phys. 9: 5131-5153

**Zhang M**.,Xu Y., Uno I., Akimoto H., 2004, "A numerical study of tropospheric ozone in the springtime in east Asia". Adv. Atmos. Sci. 21: 163-170



**Zhang L**., Brook J. R., Vet R., 2003, "A revised parameterization for gaseous dry deposition
in air-quality models". Atmos. Chem. Phys. 3: 2067-2082
**Zhu Z**., Piao S., Myneni R.B., Huang M., Zeng Z., et al., 2016, "Greening of the Earth and its
drivers". Nature Climate Change 6: 791-795



**Figure 1:** Surface ozone mean concentrations (in ppb) at the lower model layer for each ACCMIP model for the historical run and for RCP2.6, RCP4.5 and RCP8.5 simulations by 2100.

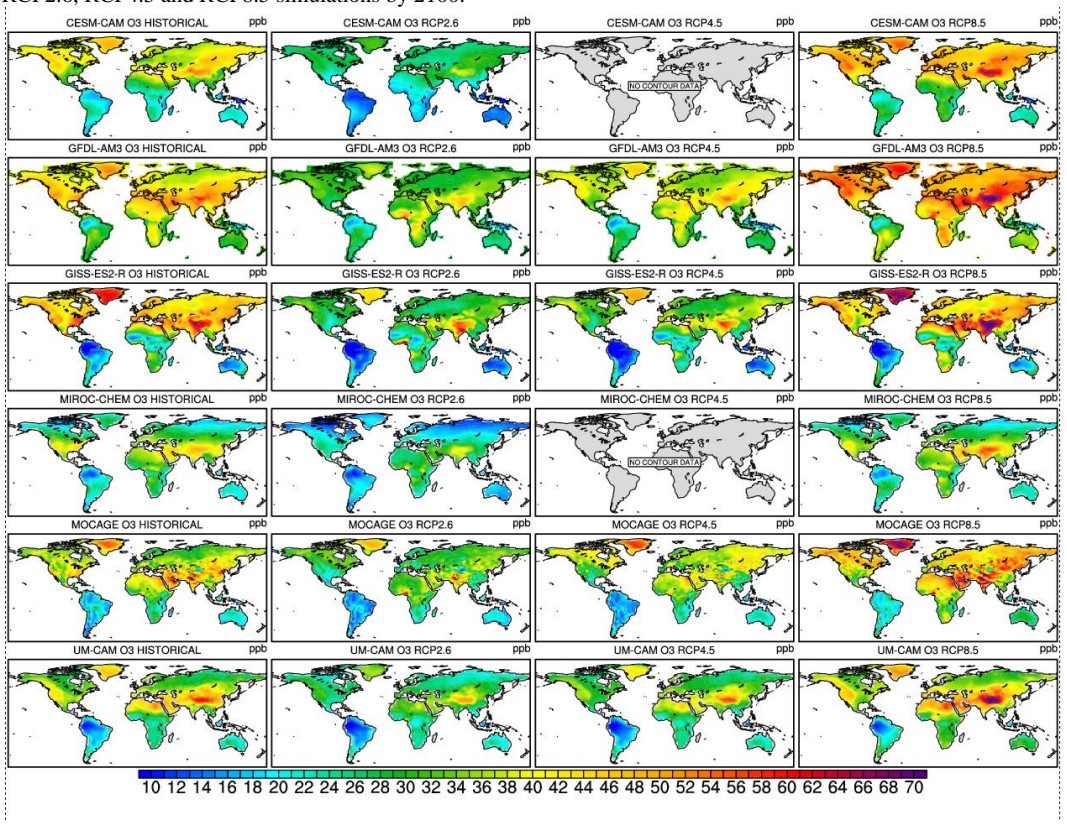



**Figure 2:** Surface AOT40 means (in ppm.h) at the lower model layer for each ACCMIP model for the historical run and for
RCP2.6, RCP4.5 and RCP8.5 simulations by 2100.

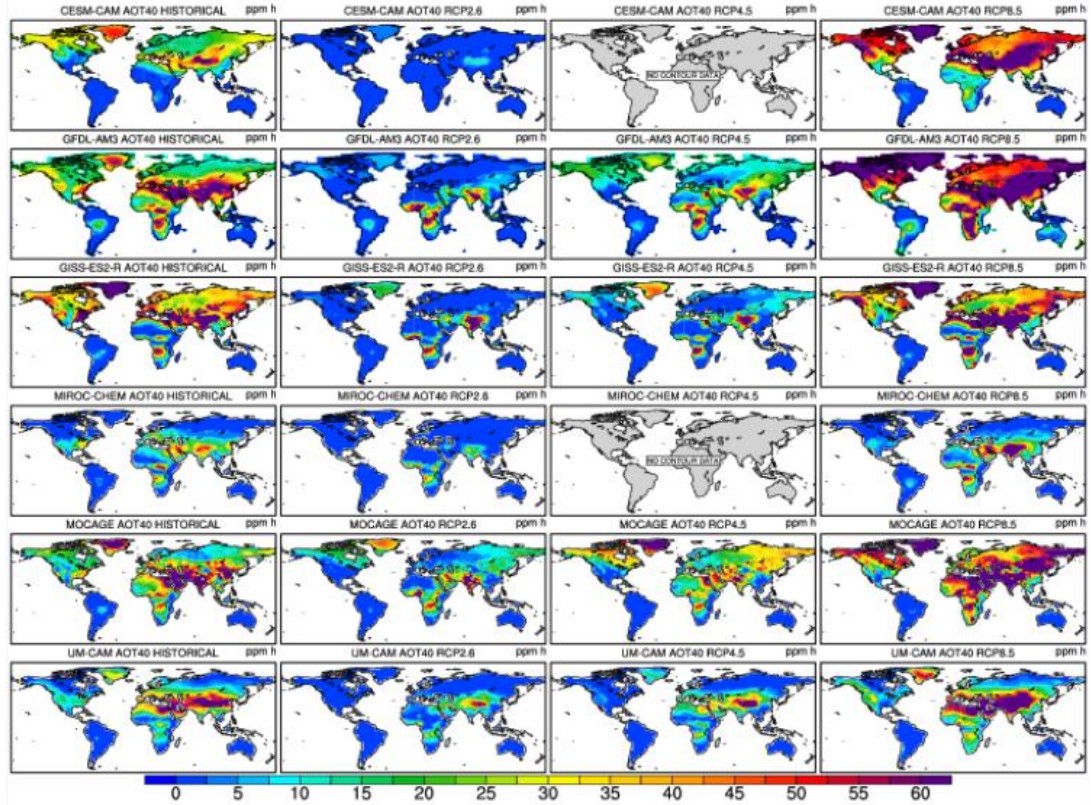






**Figure 3:** Simulated percentage changes (%) in the potential ozone impact on vegetation (IO3) for each ACCMIP model between
RCP2.6, RCP4.5 and RCP8.5 simulations and the historical run.

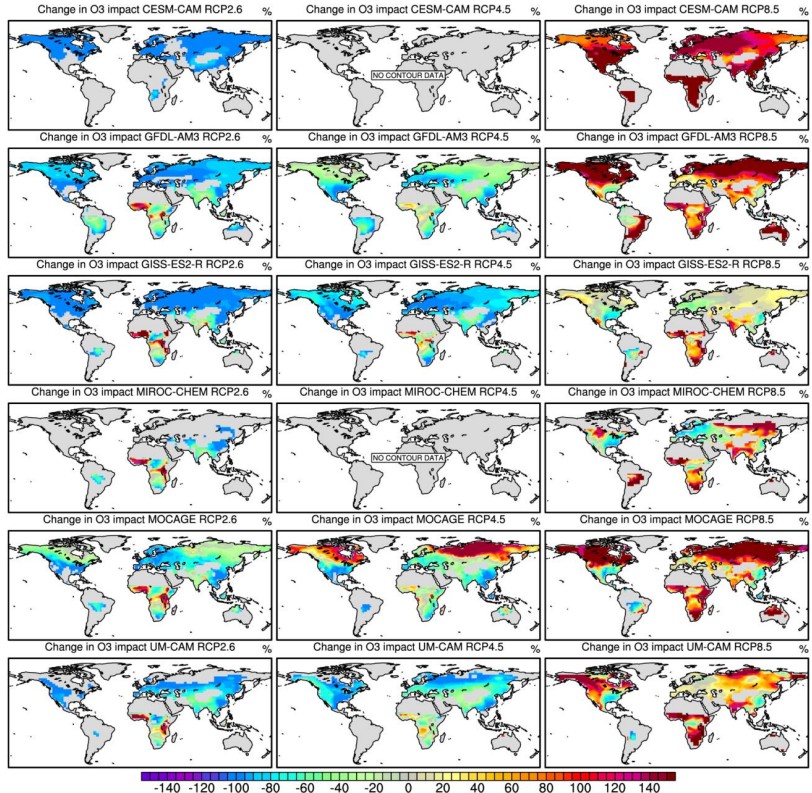


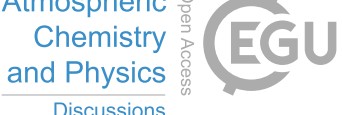



**Table 1:** Characteristics of the models, including simulation time slice, spatial resolution, simulated gas species and associated
bibliographic references (from Lamarque et al., 2013 and Young et al. 2013). Black carbon (BC), Organic carbon (OC), Secondary
Organic Aerosols (SOA), Dimethylsulfide (DMS), Chemistry Climate Model (CCM), Chemistry Transport Model (CTM),
Chemistry-General Circulation Model (CGCM).

| Models | Type | Simulation length | Resolution (lat/lon) | Number of vertical pressure levels & top level | Species simulated | References |
|--------|------|-------------------|----------------------|------------------------------------------------|-------------------|------------|
| CESM-CAM | CCM | 2000-2009 and 2100-2109 | 1.875/2.5 | 26 levels 3.5 hPa | **16 gas species**; constant present-day isoprene, soil NOx, DMS and volcanic sulfur, oceanic CO. | Lamarque et al., 2012 |
| GFDL-AM3 | CCM | 2001-2010 and 2101-2110 | 2.0/2.5 | 48 levels 0.017 hPa | **81 gas species**; SOx, BC, OC, SOA, $NH_3$, $NO_3$; constant pre-industrial soil NOx; constant present-day soil and oceanic CO, and biogenic VOC; climate-sensitive dust, sea salt, and DMS. | Donner et al., 2011 Naik et al., 2012 |
| GISS-E2-R | CCM | 2000-2004 and 2101-2105 | 2.0/2.5 | 40 levels 0.14 hPa | **51 gas species**; interactive sulfate, BC, OC, sea salt, dust, $NO_3$, SOA, alkenes; constant present-day soil NOx; climate-sensitive dust, sea salt, and DMS; climate-sensitive isoprene based on present-day vegetation. | Lee and Adams, 2011 Shindell et al., 2012 |
| MIROC-CHEM | CCM | 2000-2010 and 2100-2104 | 2.8/2.8 | 80 levels 0.003 hPa | **58 gas species**; $SO_4$, BC, OC; constant present-day VOCs, soil-NOx, oceanic-CO; climate-sensitive dust, sea salt and DMS. | Watanabe et al., 2011 |
| MOCAGE | CTM | 2000-2003 and 2100-2103 | 2.0/2.0 | 47 levels 6.9 hPa | **110 gas species**; constant present-day isoprene, other VOCs, oceanic CO and soil NOx. | Josse et al., 2004 Krinner et al., 2005 Teyssèdre et al., 2007 |
| UM-CAM | CGCM | 2000-2005 and 2094-2099 | 2.50/3.75 | 19 levels 4.6 hPa | **60 gas species**; constant present-day biogenic isoprene, soil NOx, biogenic and oceanic CO. | Zeng et al., 2008, 2010 |




**Table 2a:** Annual total emissions of CO (Tg CO/year), NMVOCs (Tg C/year), NOx (Tg N/year, including lightning and soil NOx),
total lightning NOx emissions (LNOx) and global atmospheric methane ($CH_4$) burden (Tg) for the historical simulations in each
model (from Young et al., 2013 and * from Voulgarakis et al., 2013).

| Models | Historical | | | | |
|---|---|---|---|---|---|
| | CO | * $CH_4$ | NMVOCs | NOx | *LNOx |
| CESM-CAM | 1248 | 4902 | 429 | 50.0 | 4.2 |
| GFDL-AM3 | 1246 | 4809 | 830 | 46.2 | 4.4 |
| GISS-E2-R | 1070 | 4793 | 830 | 48.6 | 7.7 |
| MIROC-CHEM | 1064 | 4805 | 833 | 57.3 | 9.7 |
| MOCAGE | 1168 | 4678 | 1059 | 47.9 | 5.2 |
| UM-CAM | 1148 | 4879 | 535 | 49.2 | 5.1 |

**Table 2b:** Simulated percentage (%) changes in total emissions of CO, NMVOCs, NOx (including lightning and soil NOx), total
lightning NOx emissions (LNOx) and global atmospheric $CH_4$ burden for each model between 2100 and historical simulation for
RCPs (from Young et al., 2013 and *Voulgarakis et al., 2013). The last row shows means and standard deviations (SD). Missing or
not available data are identified (n.a).

| Models | RCP2.6 scenario | | | | | RCP4.5 scenario | | | | | RCP8.5 scenario | | | | |
|---|---|---|---|---|---|---|---|---|---|---|---|---|---|---|---|
| | CO | *$CH_4$ | VOCs | NOx | *LNOx | CO | *$CH_4$ | VOCs | NOx | *LNOx | CO | *$CH_4$ | VOCs | NOx | *LNOx |
| CESM-CAM | - 36.7 | - 27.1 | 0 | - 52.8 | + 7.1 | n.a | n.a | n.a | n.a | n.a | - 30.1 | + 112.1 | 0 | - 33.0 | + 29.7 |
| GFDL-AM3 | - 36.9 | - 27.9 | - 5.0 | - 47.0 | + 12.6 | - 47.4 | - 9.3 | - 3.6 | - 41.5 | + 23.5 | - 30.3 | + 116.1 | - 1.9 | - 22.4 | + 38.2 |
| GISS-E2-R | - 42.8 | - 21.0 | + 0.5 | - 44.2 | + 3.8 | - 54.9 | + 4.6 | + 6.9 | - 39.2 | + 12.2 | - 35.1 | + 152.7 | + 19.8 | - 20.0 | + 26.2 |
| MIROC-CHEM | - 43.1 | - 28.2 | - 7.1 | - 36.0 | + 7.5 | n.a | n.a | n.a | n.a | n.a | - 35.4 | + 116.0 | - 3.4 | - 6.9 | + 38.0 |
| MOCAGE | - 39.4 | - 28.8 | - 6.5 | - 45.7 | + 5.2 | n.a | n.a | n.a | n.a | n.a | - 32.3 | + 113.4 | - 2.8 | - 22.9 | + 19.9 |
| UM-CAM | - 39.0 | - 27.9 | - 11.3 | - 40.6 | + 8.1 | - 50.4 | - 8.7 | - 9.2 | - 36.0 | + 17.5 | - 32.0 | + 112.1 | - 4.2 | - 17.2 | + 43.6 |
| Mean ± SD | - 39.7 ± 2.2 | - 26.8 ± 3.7 | - 4.9 ± 4.9 | - 44.4 ± 4.3 | + 7.4 ± 2.0 | - 50.9 ± 3.2 | - 4.5 ± 9.4 | - 2.0 ± 11.4 | - 38.9 ± 2.3 | + 17.7 ± 3.7 | - 32.5 ± 1.8 | + 120.4 ± 19.5 | + 1.3 ± 11.6 | - 20.4 ± 7.0 | + 32.6 ± 10.8 |




**Table 3a:** Global and hemispheric (averaged over the domain) surface ozone mean concentrations (in ppb) and AOT40 means (in ppm.h) for the historical simulations in each model (North and South Hemisphere, i.e NH and SH). The last row shows means and standard deviations (SD).

| Models | Ozone conc. global | Ozone conc. SH | Ozone conc. NH | AOT40 global | AOT40 SH | AOT40 NH |
|---|---|---|---|---|---|---|
| CESM-CAM | 31.3 | 20.9 | 36.4 | 12.8 | 0.2 | 18.9 |
| GFDL-AM3 | 38.6 | 30.6 | 42.9 | 21.8 | 4.7 | 30.8 |
| GISS-E2-R | 35.8 | 22.3 | 42.3 | 26.0 | 3.6 | 36.8 |
| MIROC-CHEM | 27.9 | 20.4 | 31.4 | 7.3 | 1.9 | 9.8 |
| MOCAGE | 32.9 | 21.5 | 38.3 | 22.9 | 3.5 | 31.8 |
| UM-CAM | 31.3 | 21.4 | 36.0 | 14.4 | 1.3 | 20.6 |
| Mean ± SD | 33.0 ± 3.8 | 22.9 ± 3.8 | 37.9 ± 4.3 | 17.5 ± 7.2 | 2.5 ± 1.7 | 24.8 ± 10.1 |

**Table 3b:** Simulated percentage (%) changes in global and hemispheric surface ozone mean concentrations and in global mean stratospheric ozone column (* from Voulgarakis et al., 2013) for each model between 2100 and historical simulation for RCPs (North and South Hemisphere, i.e NH and SH). The last row shows means and standard deviations (SD). Missing or not available data are identified (n.a).

| Models | Surface ozone mean concentrations | | | | | | | | | * Stratospheric ozone | | |
|---|---|---|---|---|---|---|---|---|---|---|---|---|
| | RCP2.6 global | RCP2.6 SH | RCP2.6 NH | RCP4.5 global | RCP4.5 SH | RCP4.5 NH | RCP8.5 global | RCP8.5 SH | RCP8.5 NH | RCP2.6 global | RCP4.5 global | RCP8.5 global |
| CESM-CAM | - 29.1 | - 20.6 | - 31.3 | n.a | n.a | n.a | + 21.9 | + 22.5 | + 20.5 | n.a | n.a | + 5.3 |
| GFDL-AM3 | - 20.5 | - 10.8 | - 24.5 | - 11.7 | - 6.9 | - 13.5 | + 15.5 | + 18.6 | + 14.5 | + 3.3 | + 3.9 | + 8.4 |
| GISS-E2-R | - 23.5 | - 5.8 | - 27.9 | - 20.4 | - 6.3 | - 23.9 | + 7.0 | + 19.3 | + 3.8 | + 8.0 | + 8.8 | + 15.1 |
| MIROC-CHEM | - 23.3 | - 12.3 | - 26.8 | n.a | n.a | n.a | + 3.9 | + 10.3 | + 2.2 | + 2.6 | n.a | + 4.2 |
| MOCAGE | - 12.8 | + 7.4 | - 18.5 | - 1.8 | + 17.7 | - 7.0 | + 23.1 | + 40.4 | + 16.7 | + 19.9 | n.a | + 23.6 |
| UM-CAM | - 17.3 | - 4.7 | - 21.1 | - 8.3 | + 0.9 | - 10.8 | + 14.4 | + 24.3 | + 11.4 | + 6.7 | + 6.9 | + 7.4 |
| Mean ± SD | - 21.1 ± 5.6 | - 7.8 ± 9.4 | - 25.0 ± 4.7 | - 10.5 ± 7.7 | + 1.4 ± 11.5 | - 13.8 ± 7.2 | + 13.8 ± 7.1 | + 22.6 ± 10.0 | + 11.5 ± 7.3 | + 8.1 ± 7.0 | + 6.5 ± 2.5 | + 10.7 ± 7.4 |





**Table 3c:** Simulated percentage (%) changes in global and hemispheric AOT40 means for each model between 2100 and historical simulation for RCPs (North and South Hemisphere, i.e NH and SH). Missing or not available data are identified (n.a).

| Models | AOT40 | | | | | | | | |
|---|---|---|---|---|---|---|---|---|---|
| | RCP2.6 global | RCP2.6 SH | RCP2.6 NH | RCP4.5 global | RCP4.5 SH | RCP4.5 NH | RCP8.5 global | RCP8.5 SH | RCP8.5 NH |
| CESM-CAM | - 96.9 | - 99.9 | - 96.8 | n.a | n.a | n.a | + 138.3 | + 150.0 | + 134.9 |
| GFDL-AM3 | - 75.2 | - 25.5 | - 78.9 | - 53.2 | - 36.2 | - 54.5 | + 96.3 | + 242.5 | + 85.1 |
| GISS-E2-R | - 78.1 | - 13.9 | - 81.2 | - 75.0 | - 27.8 | - 77.2 | + 22.3 | + 83.3 | + 19.5 |
| MIROC-CHEM | - 74.0 | - 10.5 | - 80.6 | n.a | n.a | n.a | + 20.5 | + 78.9 | + 16.3 |
| MOCAGE | - 53.7 | + 68.6 | - 59.7 | - 17.5 | + 202.9 | - 28.3 | + 85.1 | + 448.6 | + 67.0 |
| UM-CAM | - 73.6 | + 92.3 | - 76.7 | - 52.8 | +7.7 | - 54.8 | + 49.3 | + 176.9 | + 45.1 |
| Mean ± SD | - 75.2 ± 13.7 | + 1.9 ± 69.5 | - 79.0 ± 11.8 | - 49.6 ± 23.8 | + 36.6 ± 112.4 | - 53.7 ± 20.0 | + 68.6 ± 46.3 | + 196.7 ± 137.7 | + 61.3 ± 44.8 |

**Table 3d:** Simulated percentage (%) changes in potential $O_3$ impact on vegetation (IO3) for each model between 2100 and historical simulation for RCPs (North and South Hemisphere, i.e NH and SH).Missing or not available data are identified (n.a).

| Models | Risk factor IO3 | | | | | | | | |
|---|---|---|---|---|---|---|---|---|---|
| | RCP2.6 global | RCP2.6 SH | RCP2.6 NH | RCP4.5 global | RCP4.5 SH | RCP4.5 NH | RCP8.5 global | RCP8.5 SH | RCP8.5 NH |
| CESM-CAM | - 97.2 | - 91.8 | -97.5 | n.a | n.a | n.a | + 129.6 | +146.8 | +127.5 |
| GFDL-AM3 | - 69.4 | - 49.1 | - 74.8 | - 50.1 | - 61.1 | - 47.2 | + 91.9 | +95.5 | +90.4 |
| GISS-E2-R | - 66.1 | - 20.7 | - 74.3 | - 71.7 | - 53.3 | - 74.6 | + 21.5 | +56.6 | +14.2 |
| MIROC-CHEM | - 41.4 | - 18.9 | - 51.9 | n.a | n.a | n.a | + 41.0 | +103.8 | +25.5 |
| MOCAGE | - 46.6 | -22.8 | - 51.4 | - 7.0 | - 38.0 | - 1.0 | + 77.7 | +68.2 | +80.0 |
| UM-CAM | - 45.8 | - 9.2 | - 71.3 | - 59.5 | + 2.0 | - 69.0 | + 61.3 | +84.2 | +56.0 |
| Mean ± SD | - 61.1 ± 21.1 | - 35.5 ± 30.7 | - 70.2 ± 17.2 | - 47.1 ± 28.1 | - 37.6 ± 28.1 | - 47.9 ± 33.4 | + 70.5 ± 38.4 | + 92.5 ± 31.7 | + 65.6 ± 42.4 |