# Peer review of "Projected global tropospheric ozone impacts on vegetation under different 2 emission and climate scenarios"

_Atmospheric Chemistry and Physics, 2017_

## Referee Comment (RC1) · Anonymous Referee #1 · 30 Mar 2017

The work presented in this paper is based on the surface ozone concentration fields from 6 global models that have simulated atmospheric composition for recent history (around year 2000) and for the end of the 21st century under three of the IPCC future climate scenarios RCP2.6, RCP4.5 and RCP8.5. The model simulations were conducted already under the ACCMIP project. From the surface ozone concentrations the authors derive AOT40 values for ozone for each model and each simulation and also a potential vegetation injury risk factor due to ozone (IO3) for each model and each simulation. For each model, the % change in the AOT40 and IO3 values are presented for the three future scenarios relative to the historical scenario. The variations in the values of the metrics and in the simulated future changes in the metrics are discussed

in terms of differences between the 6 models, global geographic patterns, and differences between the future scenarios.

Scientific comments:

(1) The main conclusion of the work as stated in the last sentence of the abstract (and in lines 585-587) is the recommendation that improved evaluation of regional exposure of ecosystems to O3 requires improved chemistry-climate modelling systems, fully coupled with dynamic vegetation models. This is a conclusion that (a) provides no additional insight to the reader – it could have been written down before reading this study, and (b) is not based on data provided by this study – the authors do not demonstrate in this study that these modelling improvements improve the modelling of O3 ecosystem impacts.

(2) The core of this paper is the calculation and use of the AOT40 value but the description in the Methods section of how AOT40 values are calculated in this work is currently very unclear (line 143 and onwards). Was the calculation of an AOT40 for a model grid for all hours in the year, or for hours between 08.00 and 20.00 for all days of the year, or for the local 'daylight' period for all days of the year? At one point the text refers to calculation during 'daylight hours' (Line 144) but in another place 'daylight' is defined as 08.00-20.00 (Line 137), and the formula presented in Equation 1 implies calculation using all hours in the year. Derived values of AOT40 depend on this issue.

If using a 08.00-20.00 time-stamp to define daylight, the authors should confirm that this is reference to the local time for that grid cell.

If using local daylight hours then the authors should confirm how this was defined as function of latitude and day-of-year.

(3) As noted above, it appears that in this work the AOT40 value is evaluated for all days of the year, whilst, as the authors have noted, for application of AOT40 to evaluate potential vegetation damage the AOT40 value needs to be evaluated over a certain

period only during the year, a period which is different for different vegetation types. The authors state that calculating AOT40 for all days of the year in this work is not an issue when they are considering changes (in AOT40) between historic to future simulations. But surely this is not true. The seasonal distribution of ozone concentrations will change between different scenarios so relative changes in AOT40 computed for all-year will very likely be different from the relative changes in AOT40 computed for a sub-set of the year, as AOT40 values for vegetation damage assessment should be calculated. This issue needs much more discussion and justification.

(4) Having stated in the Methods section that even if they overestimate AOT40 their study is focused on the relative changes in AOT40, they then later make statements about extent of exceedance of absolute AOT40 critical values. For example line 547 in the Conclusions states: "[The] most important results from the study are the significant overrun of exposure metric (AOT40) in comparison with the AOT40-based critical level for the protection of forests (5 ppm.h) and crops (3 ppm.h). Furthermore, they appear to fail to acknowledge or take account that the AOT40 critical values for forests and crops require calculation of AOT40 over defined months, not the full year as their method in Equation 1 has done.

(5) The authors apply an alpha factor to their (all-year) AOT40 values to calculate a potential ozone vegetation risk factor IO3 (Equation 2 in Line 167). The units of alpha are quoted as per mm per ppb. When alpha is multiplied by an AOT40 value, which has units of ppb.h, this means that the IO3 metric has units of h/mm (i.e. dimensions of time per length). Can the authors explain the physical/biological basis for a photosynthetic assimilation risk factor having these dimensions?

When I check the associated citation (Anav et al. 2011, GCB) I note that the equivalent formula in this latter paper also includes a stomatal conductance variable g, which is not present in Equation 2 and not mentioned in the current manuscript.

(6) The conclusions section is almost 3 pages long and much of it is discussion/statement of prior literature and not conclusions from this work. For example, lines 552-560, lines 573-582 and lines 592-598 are generally re-statements of previous published findings and conclusions, not conclusions from this work.

(7) The abstract contains contradictory text. The first sentence states that concentrations of surface O3 are expected to increase in the future. Later in the abstract it is stated that for two of the RCP scenarios investigated ozone concentrations and vegetation injury decreases in the future.

(8) Overall, whilst the extensive discussion of the variation in surface O3 mixing ratios (geographically, with model, and with scenario) is valid (but probably also described in other publications that have emanated from the ACCMIP project), I am not convinced that statements made about changes to ozone ecosystem injury are quantitatively valid.

Minor technical/typographical corrections:

Line 17: change "overrun" to "exceedance".

Line 39: no capital letter for "globe"

Line 44: the authors refer to "tropospheric O3 levels", but do they mean "surface O3 levels" here? Throughout the Introduction there appears to be use of "tropospheric" when "surface" is meant. Ensure that terminology is used accurately.

Line 61: the authors refer to "expected" mean concentrations of O3 of 97 ppb in 2100, without comment that this is just an estimate under a particular scenario, nor comment that this increase of 50 ppb O3 in the 21st century is substantially at odds with what they write in their abstract of possible increases of a few ppb O3 under the 'worst case' RCP scenario evaluated.

Line 71: only one of the phrases "state-of-the-art" or "up-do-date" is needed here.

Line 83: the Latin "in fine" is not generally used in English, replace with something like

"and finally"

Line 84: rephrase as "This rising CO2 reduces. . ."

Lines 190-192: Please clarify whether the O3 levels quoted for the extratropics are average over land surfaces only (as was the case for the data quoted in the previous sentence) or for all extratropic areas. Similarly for the data presented in the subsequent sentence relating to AOT40 hours. It may be helpful to have a general statement somewhere that all spatial averages are for land and non-land surfaces unless stated otherwise (or whatever is an appropriate equivalent statement for this paper).

Line 220: rephrase as "the different chemical schemes used."

Lines 262-266: this paragraph contains general statements about ozone deposition that are not directly related to either the preceding or subsequent paragraphs that contain discussion of modelled O3 level in specific regions of the world.

Line 271: rephrase as "Several investigations of. . ." and delete the commas after "investigation" and after "Greenland".

Line 275: lower case for "peroxyacetyl"

Line 279: space required before "concentrations"

Line 365: write as "all the RCP scenarios include"

Line 375: change "Inversely" to "In contrast"

Line 459: rephrase as "are not associated with an increase"

Line 488-490: rephrase to make it clear whether the stated GPP reduction exceeding 30% is referring to "our results" or to the cited Sitch et al. study.

Line 515: see above for line 83.

Line 548: change "overrun" to "exceedance".

[Figure]

Caption of Figure 1: clarify exactly what is plotted "Surface ozone annual average concentrations. . ."?

Caption of Figure 2: clarify what is plotted. What is an AOT40 mean? Delete the word "mean"?

Figure 3: the middle panel on the top row has a "No contour data" error message on it.

Table 2b: Suggest moving the column with the CH4 data to be the last column in each of the groups of columns since the data in the other columns in each group refer to % change in emissions whilst the data in the CH4 column refer to % changes in atmospheric burden.

Caption of Table 3a: clarify what is presented. Is it better phrased as: "Global and hemispheric (averaged over the domain) mean annual-average surface ozone concentrations (in ppb) and mean AOT40 (in ppm.h) for. . .."?

Caption of Table 3b: clarify what is presented. Is it better phrased as: "Simulated percentage (%) changes in global and hemispheric mean annual-average surface ozone concentrations. . ."?

Caption of Table 3c: clarify what is presented. Is it better phrased as: "Simulated percentage (%) changes in global and hemispheric mean AOT40. . ."?

---

## Author Comment (AC1) · 22 Apr 2017

Dear Reviewer,

Thank you for giving us the opportunity of a reply. We hope that we have satisfactorily addressed all their concerns.

Best regards

**Pierre Sicard**

Bullet (1) - The main conclusion of the work as stated in the last sentence of the abstract (and in lines 585-587) is the recommendation that improved evaluation of regional exposure of ecosystems to O3 requires improved chemistry-climate modelling systems, fully coupled with dynamic vegetation models. This is a conclusion that (a) provides no additional insight to the reader – it could have been written down before reading this study, and (b) is not based on data provided by this study – the authors do not demonstrate in this study that these modelling improvements improve the modelling of O3 ecosystem impacts.

Reply - The reviewer is right. We will remove this sentence in the abstract and reformulate the section "Conclusions".

Bullet (2) - The core of this paper is the calculation and use of the AOT40 value but the description in the Methods section of how AOT40 values are calculated in this work is currently very unclear (line 143 and onwards). Was the calculation of an AOT40 for a model grid for all hours in the year, or for hours between 08.00 and 20.00 for all days of the year, or for the local 'daylight' period for all days of the year? At one point the text refers to calculation during 'daylight hours' (Line 144) but in another place 'daylight' is defined as 08.00-20.00 (Line 137), and the formula presented in Equation 1 implies calculation using all hours in the year. Derived values of AOT40 depend on this issue. If using a 08.00-20.00 time-stamp to define daylight, the authors should confirm that this is reference to the local time for that grid cell. If using local daylight hours then the authors should confirm how this was defined as function of latitude and day-of-year.

Reply - We realize that it was not clear that we computed the AOT40 for a model grid for hours between 8am and 8pm (local time) for all days of the year. We will state it clearly in the text. Conventionally, two major growing-season time windows are used, i.e. six months (April to September) for temperate climates, e.g. in Europe (CLRTAP, 2015) and all-year round for Mediterranean, subtropical and tropical-type climates where vegetation is physiologically active all along the year (Paoletti et al., 2007). UNECE (2010) supports the use of a growing season, but a fixed time-window does not allow incorporating the changes in the growing season due to climate change

**ACPD**
and would thus not be well suited when investigating changes over time. In addition, AOT40 is widely used not only in Europe (e.g. Anav et al., 2016; De Marco et al., 2016) but also in South America (Moura et al., 2014) and Asia (Hoshika et al., 2011).

The use of the fixed time-window 8-20 (as defined by the Directive 2008/50/EC) allyear round allows skipping the use of a latitude model, which would increase the level of complexity and uncertainties. We believe that this approach is valuable as it can be easily applied at global scale. We will include this justification in the text.

Bullet (3) - As noted above, it appears that in this work the AOT40 value is evaluated for all days of the year, whilst, as the authors have noted, for application of AOT40 to evaluate potential vegetation damage the AOT40 value needs to be evaluated over a certain period only during the year, a period which is different for different vegetation types. The authors state that calculating AOT40 for all days of the year in this work is not an issue when they are considering changes (in AOT40) between historic to future simulations. But surely this is not true. The seasonal distribution of ozone concentrations will change between different from the relative changes in AOT40 computed for a sub-set of the year, as AOT40 values for vegetation damage assessment should be calculated. This issue needs much more discussion and justification.

Reply - Selecting a common time window at global level is an issue because the growing season is highly variable across the latitude. Rather than introducing further uncertainties by using a latitude model to simulate the growing season, we applied here a simplified approach with a year-long growing season which should be considered as a worst case study. This way, we were able to compare the historical and projected potential risk to vegetation. We will introduce a note of caution about the limitations of the present study.

Bullet (4) - Having stated in the Methods section that even if they overestimate AOT40 their study is focused on the relative changes in AOT40, they then later make state-

ACPD
ments about extent of exceedance of absolute AOT40 critical values. For example line 547 in the Conclusions states: "[The] most important results from the study are the significant overrun of exposure metric (AOT40) in comparison with the AOT40-based critical level for the protection of forests (5 ppm.h) and crops (3 ppm.h). Furthermore, they appear to fail to acknowledge or take account that the AOT40 critical values for forests and crops require calculation of AOT40 over defined months, not the full year as their method in Equation 1 has done.

Reply - We would like to thank the reviewer for this comment. We will introduce a note of caution when discussing the AOT40-based critical levels for the protection of forests and crops. Most important results from the study are the spatial pattern and projected changes in global AOT40 and risk areas for vegetation under the RCP scenarios. This is a very novel result of our study.

Bullet (5) - The authors apply an alpha factor to their (all-year) AOT40 values to calculate a potential ozone vegetation risk factor IO3 (Equation 2 in Line 167). The units of alpha are quoted as per mm per ppb. When alpha is multiplied by an AOT40 value, which has units of ppb.h, this means that the IO3 metric has units of h/mm (i.e. dimensions of time per length). Can the authors explain the physical/biological basis for a photosynthetic assimilation risk factor having these dimensions? When I check the associated citation (Anav et al. 2011, GCB) I note that the equivalent formula in this latter paper also includes a stomatal conductance variable g, which is not present in Equation 2 and not mentioned in the current manuscript.

Reply - We would like to thank the reviewer for this comment. The potential O3 impact on photosynthetic assimilation (IO3) is expressed through a dimensionless value. As for the variable g (i.e. stomatal conductance), ACCMIP models do not provide this variable as output, thus we can only compute the likely impact of O3 as the product of the sensitivity coefficient and the O3 concentration. We consider this impact as a potential one, in the worst-case scenario, where all the ozone is entering into the leaf. ACPD
After a deep review of both papers, i.e. Reich (1987) and Ollinger et al. (1997),  $\alpha$  is an empirically derived O3 response coefficient (dimensionless value) representing the proportional change in photosynthesis and biomass growth per unit of ozone-uptake. IO3 is the simulated percentage changes (%) in the potential ozone injury on vegetation between that expected at the end of the 21st century (RCPs simulations) and present.

 $\alpha$  × AOT40 in ppb h IO3 =  $\alpha$  × (AOT4021st century - AOT40 present) / (AOT40 present) x 100=> IO3 in %

A statement will be added to clarify this issue and the units will be modified.

Bullet (6) - The conclusions section is almost 3 pages long and much of it is discussion/statement of prior literature and not conclusions from this work. For example, lines 552-560, lines 573-582 and lines 592-598 are generally re-statements of previous published findings and conclusions, not conclusions from this work.

Reply - We will shorten the section "Conclusions" taking into account the referee comments.

Bullet (7) - The abstract contains contradictory text. The first sentence states that concentrations of surface O3 are expected to increase in the future. Later in the abstract it is stated that for two of the RCP scenarios investigated ozone concentrations and vegetation injury decreases in the future.

Reply - We will reformulate the abstract.

Bullet (8) - Overall, whilst the extensive discussion of the variation in surface O3 mixing ratios (geographically, with model, and with scenario) is valid (but probably also described in other publications that have emanated from the ACCMIP project), I am not convinced that statements made about changes to ozone ecosystem injury are quantitatively valid.

Reply - It is not the purpose of this study to offer a quantitative estimation of the ecosystem injury due to ozone but to highlight the world areas at higher risk in a worst case

**ACPD**
scenario, and how they change relative to the historical situation. A statement will be added to clarify this issue.

Bullet (9) - Minor technical/typographical corrections:

All requested technical and typographical corrections were carried out. We have mentioned "No contour data" in the panel for two models for which the data were missing under RCP4.5.

---

## Referee Comment (RC2) · Anonymous Referee #3 · 10 Jul 2017

-General comments This is an interesting scientific study to evaluate the future impacts of ozone on vegetation. The authors tested the different RCP scenarios and compared results by the six global chemistry models. The topic is very actual and the study is carefully done. The results are an important basis for further developments on future ozone risks for global vegetation. There are only a few minor remarks.

-Specific comments

L142 "when the stomatal conductance is greater than 0": what do you mean? Do you mean the "leafy season"? Please rephrase it.

L147 "the overestimation of AOT40 does not affect our results": it is not clear why.

[Figure]

Please rephrase it.

L170 not "per unit of ozone-uptake" but "per unit of AOT40"

L172 Again you did not use "ozone-uptake" in Eq. (2). You can describe it as "regressions of the photosynthesis response to ozone (Reich, 1987)".

L173 What are the "other vegetation types"? And please justify why the photosynthetic responses to AOT40 are same between deciduous trees and "other vegetation types".

L464-465 Nemani et al. (2003) and Zhu et al. (2016) did not show the ozone impacts. Please revise it.

L480-481 "In these areas, the increasing effect of a warming. . .": where can we refer for this result? Please specify it.

L491-496 "mainly due to the lack of empirical data about the response of different species to O3": We have to say that this is a weak rationale. In fact, Sitch et al. (2007) considered five plant types (broad-leaved tree, needle-leaved tree, C3 crops, C4 crops and shrubs; please see the Table S1 of their paper). But we can find a marked difference in estimated ozone concentration in 2100 between this study (Fig. 1) and Sitch et al. (2007). A major advantage of this study is a comparison between the models and scenarios. The authors should reconsider the sentence and should emphasize what is the need to explore future potential impacts of ozone.

L553-560: I agree with the statement. However, if so, readers are wondering why AOT40 was targeted in this paper. The authors can put more "take-home messages" for readers. For example, what is a climatic condition (arid/humid) in high AOT40 regions? How about the need for a parameterization of the ozone dose-response relationships in tropical plants? ...etc.

L578 "the lower risk areas include evergreen broadleaf forests ": we cannot find the description about the parameters in evergreen broadleaf forests (lines 170-174). Did you target this plant type?

Figure 3 legend: "the potential ozone impact on vegetation": of what? Maybe photo-synthetic assimilation. But please specify it.

-Technical corrections

L551 "..South Asia they may..": you had better put ", and" before "they may".

L552 not "were" but "was"

---

## Author Response (AR1)

Dear Reviewers,

Thank you for giving us the opportunity of a reply. We hope that we have satisfactorily addressed all concerns.

Best regards

Pierre Sicard

**Anonymous Referee #2**

**Scientific comments:**

**(1)** The main conclusion of the work as stated in the last sentence of the abstract (and in lines 585-587) is the recommendation that improved evaluation of regional exposure of ecosystems to O3 requires improved chemistry-climate modelling systems, fully coupled with dynamic vegetation models. This is a conclusion that (a) provides no additional insight to the reader – it could have been written down before reading this study, and (b) is not based on data provided by this study – the authors do not demonstrate in this study that these modelling improvements improve the modelling of $O_3$ ecosystem impacts.

The reviewer is right. We will remove this sentence in the abstract and reformulate the section "Conclusions".

**(2)** The core of this paper is the calculation and use of the AOT40 value but the description in the Methods section of how AOT40 values are calculated in this work is currently very unclear (line 143 and onwards). Was the calculation of an AOT40 for a model grid for all hours in the year, or for hours between 08.00 and 20.00 for all days of the year, or for the local 'daylight' period for all days of the year? At one point the text refers to calculation during 'daylight hours' (Line 144) but in another place 'daylight' is defined as 08.00-20.00 (Line 137), and the formula presented in Equation 1 implies calculation using all hours in the year. Derived values of AOT40 depend on this issue. If using a 08.00-20.00 time-stamp to define daylight, the authors should confirm that this is reference to the local time for that grid cell. If using local daylight hours then the authors should confirm how this was defined as function of latitude and day-of-year.

We realize that it was not clear that we computed the AOT40 for a model grid for hours between 8am and 8pm (local time) for all days of the year. We will state it clearly in the text. Conventionally, two major growing-season time windows are used, i.e. six months (April to September) for temperate climates, e.g. in Europe (CLRTAP, 2015) and all-year round for Mediterranean, subtropical and tropical-type climates where vegetation is physiologically active all along the year (Paoletti et al., 2007). UNECE (2010) supports the use of a growing season, but a fixed time-window does not allow incorporating the changes in the growing season due to climate change and would thus not be well suited when investigating changes over time. In addition, AOT40 is widely used not only in Europe (e.g. Anav et al., 2016; De Marco et al., 2016) but also in South America (Moura et al., 2014) and Asia (Hoshika et al., 2011).

The use of the fixed time-window 8-20 (as defined by the Directive 2008/50/EC) all-year round allows skipping the use of a latitude model, which would increase the level of

complexity and uncertainties. We believe that this approach is valuable as it can be easily applied at global scale. We will include this justification in the text.

**(3)** As noted above, it appears that in this work the AOT40 value is evaluated for all days of the year, whilst, as the authors have noted, for application of AOT40 to evaluate potential vegetation damage the AOT40 value needs to be evaluated over a certain period only during the year, a period which is different for different vegetation types. The authors state that calculating AOT40 for all days of the year in this work is not an issue when they are considering changes (in AOT40) between historic to future simulations. But surely this is not true. The seasonal distribution of ozone concentrations will change between different scenarios so relative changes in AOT40 computed for all-year will very likely be different from the relative changes in AOT40 computed for a sub-set of the year, as AOT40 values for vegetation damage assessment should be calculated. This issue needs much more discussion and justification.

Selecting a common time window at global level is an issue because the growing season is highly variable across the latitude. Rather than introducing further uncertainties by using a latitude model to simulate the growing season, we applied here a simplified approach with a year-long growing season which should be considered as a worst case study. This way, we were able to compare the historical and projected potential risk to vegetation. We will introduce a note of caution about the limitations of the present study.

**(4)** Having stated in the Methods section that even if they overestimate AOT40 their study is focused on the relative changes in AOT40, they then later make statements about extent of exceedance of absolute AOT40 critical values. For example line 547 in the Conclusions states: "[The] most important results from the study are the significant overrun of exposure metric (AOT40) in comparison with the AOT40-based critical level for the protection of forests (5 ppm.h) and crops (3 ppm.h). Furthermore, they appear to fail to acknowledge or take account that the AOT40 critical values for forests and crops require calculation of AOT40 over defined months, not the full year as their method in Equation 1 has done.

We would like to thank the reviewer for this comment. We will introduce a note of caution when discussing the AOT40-based critical levels for the protection of forests and crops. Most important results from the study are the spatial pattern and projected changes in global AOT40 and risk areas for vegetation under the RCP scenarios. This is a very novel result of our study.

**(5)** The authors apply an alpha factor to their (all-year) AOT40 values to calculate a potential ozone vegetation risk factor IO3 (Equation 2 in Line 167). The units of alpha are quoted as per mm per ppb. When alpha is multiplied by an AOT40 value, which has units of ppb.h, this means that the IO3 metric has units of h/mm (i.e. dimensions of time per length). Can the authors explain the physical/biological basis for a photosynthetic assimilation risk factor having these dimensions? When I check the associated citation (Anav et al. 2011, GCB) I note that the equivalent formula in this latter paper also includes a stomatal conductance variable g, which is not present in Equation 2 and not mentioned in the current manuscript.

We would like to thank the reviewer for this comment. The potential $O_3$ impact on photosynthetic assimilation (IO3) is expressed through a dimensionless value. As for the variable g (i.e. stomatal conductance), ACCMIP models do not provide this variable as output, thus we can only compute the likely impact of $O_3$ as the product of the sensitivity coefficient and the $O_3$ concentration. We consider this impact as a potential one, in the worstcase scenario, where all the ozone is entering into the leaf. After a deep review of both papers, i.e. Reich (1987) and Ollinger et al. (1997), is an empirically derived $O_3$ response coefficient (dimensionless value) representing the proportional change in photosynthesis and biomass growth per unit of AOT40. IO3 is the simulated percentage changes (%) in the potential ozone injury on vegetation between that expected at the end of the 21[st] century (RCPs simulations) and present. A statement will be added to clarify this issue and the units will be modified.

$$IO3 = \alpha \times AOT40 => IO3 \text{ in ppb h}$$
$$IO3 = \alpha \times (AOT40_{21st\ century} - AOT40_{present}) / (AOT40_{present}) \times 100 => IO3 \text{ in \%}$$

**(6)** The conclusions section is almost 3 pages long and much of it is discussion/statement of prior literature and not conclusions from this work. For example, lines 552-560, lines 573-582 and lines 592-598 are generally re-statements of previous published findings and conclusions, not conclusions from this work.

We will shorten the section "Conclusions" taking into account the referee comments.

**(7)** The abstract contains contradictory text. The first sentence states that concentrations of surface O3 are expected to increase in the future. Later in the abstract it is stated that for two of the RCP scenarios investigated ozone concentrations and vegetation injury decreases in the future.

We will reformulate the abstract.

**(8)** Overall, whilst the extensive discussion of the variation in surface $O_3$ mixing ratios (geographically, with model, and with scenario) is valid (but probably also described in other publications that have emanated from the ACCMIP project), I am not convinced that statements made about changes to ozone ecosystem injury are quantitatively valid.

It is not the purpose of this study to offer a quantitative estimation of the ecosystem injury due to ozone but to highlight the world areas at higher risk in a worst case scenario, and how they change relative to the historical situation. A statement will be added to clarify this issue.

**Minor technical/typographical corrections:**

All requested technical and typographical corrections were carried out. We have mentioned "No contour data" in the panel for two models for which the data were missing under RCP4.5.

**Anonymous Referee #3**

L142 "when the stomatal conductance is greater than 0": what do you mean? Do you mean the "leafy season"? Please rephrase it.

The reviewer is right. We realize that it was not clear that we computed the AOT40 for a model grid for hours between 8am and 8pm (local time) for all days of the year. We will state it clearly in the text and we have removed this sentence.

L147 "the overestimation of AOT40 does not affect our results": it is not clear why. Please rephrase it.

The aim of this study is to assess how $O_3$ stress to vegetation changes between historical period and future. By calculating AOT40 year-round, an overestimation can be observed over polluted region. Even if the AOT40 is misestimated at a given model grid point, as we compared the mean change between present and future at the same model grid point, thus the change is consistent. We rephrased to stress that an overestimation of AOT40 does not affect our main conclusions (instead of "results") about the percentage of change in the potential $O_3$ impact on photosynthetic assimilation.

L170 not "per unit of ozone-uptake" but "per unit of AOT40"

The reviewer is right. Alpha is an empirically derived ozone response coefficient.

L172 Again you did not use "ozone-uptake" in Eq. (2). You can describe it as "regressions of the photosynthesis response to ozone (Reich, 1987)".

The reviewer is right. Data from the literature demonstrate strong relationships between cumulative ozone exposure and reductions in both net photosynthesis and plant growth. Figures from Reich (1987) show the percentage of change of photosynthesis in relation to ozone exposure, so we reworded as suggested.

L173 What are the "other vegetation types"? And please justify why the photosynthetic responses to AOT40 are same between deciduous trees and "other vegetation types".

We would like to thank the reviewer for this comment. The photosynthetic responses to AOT40 are not the same between deciduous trees and "other vegetation types". We now clearly explained in the text that the relationships between cumulative ozone exposure and reductions in both net photosynthesis and plant growth vary among and even within species (Reich, 1987; Ollinger et al., 1997; Anav et al., 2011). Differences in response per unit uptake tend to be greater in magnitude between functional groups (e.g., hardwoods vs. conifers) where leaf structure and plant growth strategy differ most widely (Reich, 1987).

From the Global Land Cover Facility (GLCF) data at 1degree of spatial resolution, we grouped the vegetation in 3 categories and then we used the following factors: conifers, crops and deciduous trees. Ollinger et al., 1997 derived a leaf-level ozone response equation for broadleaved deciduous species (2.6 x $10^{-6}$) and we used 0.7 x $10^{-6}$ for coniferous and 3.9 x $10^{-6}$ for crops (Reich, 1987).

L464-465 Nemani et al. (2003) and Zhu et al. (2016) did not show the ozone impacts. Please revise it.

The reviewer is right. These analyses (Nemani et al., 2003; Zhu et al., 2016) focused on impacts of global environmental changes (e.g. climate, land-cover, nitrogen deposition, $CO_2$ fertilization) on vegetation. We have reworded as suggested.

L480-481 "In these areas, the increasing effect of a warming: : :": where can we refer for this result? Please specify it.

We compared the GPP reduction (from - 10 to - 20%) due to $O_3$ (Sitch et al., 2007) and the strong increase in NPP and LAI due to climate change (Nemani et al., 2003; Zhu et al., 2016)

over Amazon forest. We have reformulated as *"In these areas, we observed an increasing effect of a warming climate on forests (e.g. increase in greening, NPP, LAI) as compared to a reduction in GPP due to $O_3$ (Sitch et al., 2007)"*.

L491-496 "mainly due to the lack of empirical data about the response of different species to O3": We have to say that this is a weak rationale. In fact, Sitch et al. (2007) considered five plant types (broad-leaved tree, needle-leaved tree, C3 crops, C4 crops and shrubs; please see the Table S1 of their paper). But we can find a marked difference in estimated ozone concentration in 2100 between this study (Fig. 1) and Sitch et al. (2007). A major advantage of this study is a comparison between the models and scenarios. The authors should reconsider the sentence and should emphasize what is the need to explore future potential impacts of ozone.

The reviewer is right. The ozone concentrations over Amazon forest are lower in Sitch et al., (2007), i.e. 75-90 ppb in summer (present) and more than 90 ppb by 2100. In our study, the annual $O_3$ mean is around 15-20 ppb by 2100. In this section, we added in the text explanations about the overestimation in GPP reductions simulated by Sitch et al. (2007) in summer such as the estimated $O_3$ concentration in 2100, the lack of empirical data about the response of different species to $O_3$ the non-inclusion of the nitrogen limitation of growth.

L553-560: I agree with the statement. However, if so, readers are wondering why AOT40 was targeted in this paper. The authors can put more "take-home messages" for readers. For example, what is a climatic condition (arid/humid) in high AOT40 regions? How about the need for a parameterization of the ozone dose-response relationships in tropical plants? ...etc.

The reviewer is right and we decided to add a few more information about the AOT40 limitations at global scale (e.g. factors affecting stomata e.g. water availability).

L578"the lower risk areas include evergreen broadleaf forests ": we cannot find the description about the parameters in evergreen broadleaf forests (lines 170-174). Did you target this plant type?

We did not focus on evergreen, as now clearly explained in the text. By using the land-cover data (GLCF), we can observe that the lower $O_3$ risk areas (Figure 3) correspond to areas with evergreen broadleaf forests.

Figure 3 legend: "the potential ozone impact on vegetation": of what? Maybe photosynthetic assimilation. But please specify it.

Indeed, we added "the potential O3 impact on photosynthetic assimilation".

L551 "..South Asia they may..": you had better put ", and" before "they may".

Done

L552 not "were" but "was"

Done

---

## Author Response (AR2)

Dear Reviewer,

Thank you for giving us the opportunity of a reply. We hope that we have satisfactorily addressed all queries.

Best regards

Pierre Sicard

**Anonymous Referee #2**

*The specific methodological queries have largely been addressed, although in the revised text the authors have now introduced new lack of clarity in how they assigned particular IO3 values to each model grid square.*

*In L160 it is stated that vegetation was grouped into the three categories of 'conifer', 'crops' and 'deciduous trees.' What are tropical rain forests classified as? What are grasslands and dry shrub classified as? These don't seem to map readily onto the three specified vegetation classifications.*

**R -** In this study, we have considered tropical forests and shrubs as deciduous trees while grassland was classified as cropland, similar to Sitch et al. (2007), as we have clarified this point in the text.

Even, Dynamic Global Vegetation Models make use of plant functional type rather than complex and specific vegetation to simulate shifts in potential vegetation as a response to shifts in climate. In Sitch et al (2007), data from field observation were used to calibrate plant-ozone effects for the five plant functional types described by MOSES land-surface scheme (Broadleaf trees, Needleleaf trees, C3 Grass, C4 Grass & Shrub).

*I am also not clear how a % change in IO3 can be derived if the AOT40 value in a particular grid square was initially zero (L425). An IO3 value may become non-zero in a new scenario, where previously it was zero, but it is not possible to assign a % change to a change from zero.*

**R –** The percentage of change is computed as following: [(RCPx - hist) / hist] * 100

Clearly if the AOT40 during the historical period is 0 then the percentage of change is undefined and therefore we have considered and set these grid points as missing values, as we will now state in the text. For example, readers can see there is no color in Figure 3 for the CESM-CAM model in South America (Venezuela coast) because we have an abrupt change in AOT40 between the historical run (AOT40 was 0) and RCP8.5 projections (Figure 2).

*Whilst the authors may have presented a lot of careful and numerically-correct calculations, I still wonder whether the use of various 'globally-wide' approximations means that one has to be very cautious about taking the results, particularly for changes in IO3, too literally. Example of these 'global' approximations include the calculation of AOT40 values from 08.00 to 20.00 everywhere globally regardless of latitude and time of year, the calculation of AOT40 for a full-year, and the assignment of all global vegetation to just 3 vegetation classes. The authors have responded that because they are interested in examining relative changes in impacts on vegetation these issues are not so important, but surely if growing season length (and extent of daylight hours) at a particular location is relevant for ozone-induced injury*

*then how the annual distributions of ozone concentrations change in the future under different scenarios (and also how growing seasons change under different scenarios) will have a quantitative impact on the extent of ozone-induced injury in the real world compared to these modelled worlds.*

**R** – The current chemistry models cannot predict changes in phenology, thus the growing season length is the same between the historical period and different RCPs. Here, we applied the same approximation of ACCMIP models. However, a note of caution including a citation to the phenological issue (Anav et al., 2017) will be included in the discussion.

Plant phenology plays a pivotal role in the climate system as it regulates the gas exchange between the biosphere and the atmosphere. Currently, in many risk assessment studies, the phenology function is based on a simple latitude and topography model. The Chemistry Models do not take into account the shifts in plant phenology and in start and end date of the growing season, however a first attempt to study the role of phenology on stomatal ozone uptake is shown by Anav et al (2017).

*Much of the conclusions section remains more like general discussion (particularly in its extensive reference to prior literature) than 'take home' conclusions for the reader.*

**R** – The section "conclusions" was shortened and a few references were moved towards the section "discussion" or removed.

*There are also a few issues with some of the text in this section: (i) the sentence in L529-530 is not a complete sentence; (ii) the statement in L548-549 that AOT40-based critical levels "will be exceeded over many areas…" is not clear: to which RCP scenario(s) is this sentence referring? It is also potentially a bit misleading because even though AOT40 may be exceeded in certain areas, actually for two of the three future scenarios investigated AOT40 values will be less exceeded in the future than in recent history; (iii) L569 refers to sensitivity of grasslands to ozone injury but yet the authors' methodology does not refer to how IO3 is calculated for grasslands.*

**R** – All these issues were addressed.

*Please explicitly state somewhere in the text that tabulated and quoted values of global, NH and SH mean surface ozone, AOT40, and IO3 are derived from averaging values over the global/NH/SH land areas only, not over their full respective geographic domains.*

**R** – We agree to add an explicit statement in the reviewed version for tables 3. Lines 184-186, we have addressed this point by adding this statement.

[revised manuscript text omitted]
 | - 61.1 ± 21.1 | - 35.5 ± 30.7 | - 70.2 ± 17.2 | - 47.1 ± 28.1 | - 37.6 ± 28.1 | - 47.9 ± 33.4 | + 70.5 ± 38.4 | + 92.5 ± 31.7 | + 65.6 ± 42.4 |

---

## Author Response (AR3)

Dear Editor,

Thank you for giving us the opportunity of a reply. We hope that we have satisfactorily addressed all concerns.

Best regards

Pierre Sicard
* * *
**Referee comments**

All requested technical and typographical corrections were carried out.

Line 17: yes, temperature change can enhance, but there is also a threshold for this. We cannot generalize this.

I quite agree with this comment, however the term "enhance plant growth" was used by Nemani et al. (2003) and Zhu et al. (2016) and seems appropriated.

Line 96 & Line 188: Please make a complete sentence to introduce the table in the article, not in the bracket.

We have introduced Tables and Figures, see lines 185-189

Line 134: "cannot predict" at all? Or "problems in predicting"? Give a reference too

This statement was proposed and deeply discussed by Anav et al. (Global Change Biol., 2017). Currently, the chemistry models do not take into account the shifts in plant phenology and in start and end date of the growing season.

Line 417: do these models have stratospheric chemistry?

Yes, this is explained in the SI. For stratospheric $O_3$ projections, the models are grouped into 2 categories: the first group includes models with interactive or semi-offline chemistry and the second group includes models with prescribed $O_3$. Some models (e.g. GFDL-AM3, GISS-E2-R, MIROC-CHEM and MOCAGE) include full stratospheric chemistry schemes, while CESM-CAM is based on a linearized $O_3$ chemistry (i.e. LINOZ scheme) and UM-CAM uses the CMIP5 dataset to prescribe offline $O_3$ in the stratosphere.